# kNNSampler: Stochastic Imputations for Recovering Missing Value Distributions

**Parastoo Pashmchi**                           *parastoo.pashmchi@sap.com*
*SAP Labs France E-Mobility Research*
*EURECOM, Sophia Antipolis, France*

**Jérôme Benoit**                                *jerome.benoit@sap.com*
*SAP Labs France E-Mobility Research*

**Motonobu Kanagawa**                            *motonobu.kanagawa@eurecom.fr*
*EURECOM, Sophia Antipolis, France*

**Reviewed on OpenReview:** *https://openreview.net/forum?id=4CDnIACCQG*

## Abstract

We study a missing-value imputation method, termed kNNSampler, that imputes a given unit's missing response by randomly sampling from the observed responses of the $k$ most similar units to the given unit in terms of the observed covariates. This method can sample unknown missing values from their distributions, quantify the uncertainties of missing values, and be readily used for multiple imputation. Unlike popular kNNImputer, which estimates the conditional mean of a missing response given an observed covariate, kNNSampler is theoretically shown to estimate the conditional distribution of a missing response given an observed covariate. Experiments illustrate the performance of kNNSampler. The code for kNNSampler is made publicly available.[1]

**Keywords: missing values imputation, k nearest neighbours, conditional distribution, kernel mean embedding**

## 1 Introduction

Missing values occur in real-world datasets for various reasons, such as non-response in surveys and sensor failures. Imputation — filling in missing values with their estimates — is a common preprocessing step used to address missing data. Over the decades, various imputation methods have been proposed, ranging from simple statistical techniques to machine learning algorithms (e.g., Rubin, 1976; Schafer, 1997; Schafer and Graham, 2002; Little and Rubin, 2002; Mattei and Frellsen, 2019; Enders, 2022).

kNNImputer (Troyanskaya et al., 2001) is one of the most widely used imputation methods, owing to its simplicity and availability in popular software packages such as scikit-learn[2] (Pedregosa et al., 2011). It imputes a missing response variable (e.g., customer satisfaction level) of a given unit (e.g., a customer) as the average of the observed responses of the $k$ most similar units to the given unit in terms of observed covariates (e.g., age, gender, occupation). This is to predict the missing response by $k$ nearest neighbours (kNN) regression (Stone, 1977) so the imputation is an estimate of the conditional *expectation* of the missing response given a covariate. The method has been widely used in science and engineering, and many extensions have been proposed (e.g., García-Laencina et al., 2009; Tutz and Ramzan, 2015; De Silva and Perera, 2016; Huang et al., 2017; Faisal and Tutz, 2021).

An issue of kNNImputer, shared by other regression-based imputers, is that the distribution of imputations can be significantly different from the distribution of true (hidden) missing values. This is because, as

---

[1]https://github.com/SAP/knn-sampler
[2]https://scikit-learn.org/stable/modules/generated/sklearn.impute.KNNImputer.html

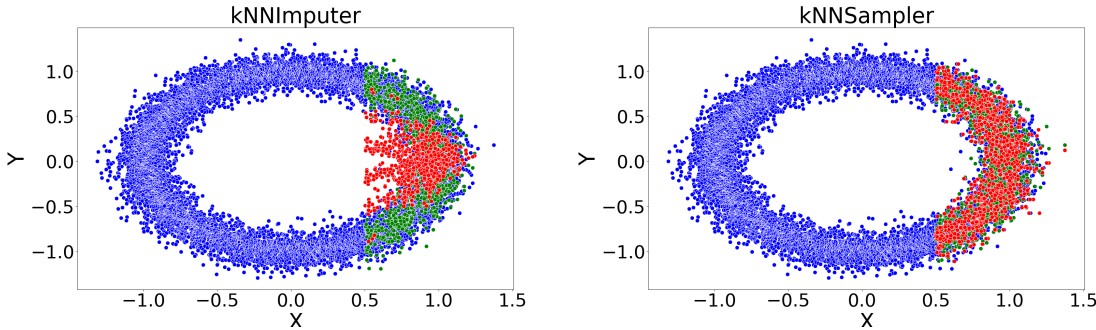

Figure 1: Comparison of imputations by kNNImputer (left) and kNNSampler (right). In each figure, $x$ and $y$ are the covariate and response, respectively. Blue points are observed covariate-response pairs, green points are true missing values and red points are imputed values. For details, see Section 4.

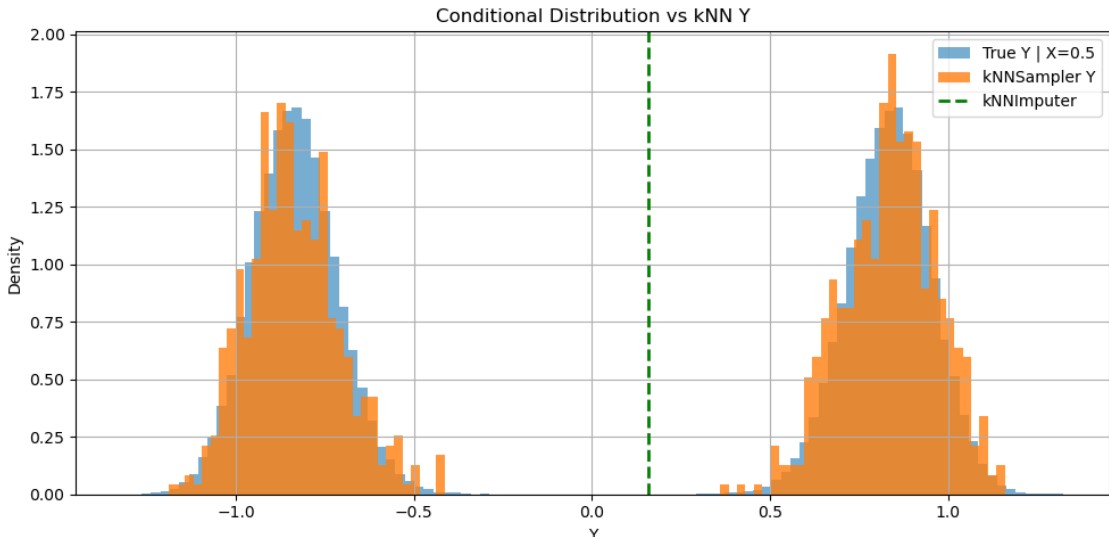

Figure 2: Comparison of the samples of the true conditional distribution $P(y|x)$ of missing response $y$ of a unit with covariate $x = 0.5$ (blue) and the kNN conditional distribution $\hat{P}(y|x)$ with $k = 1,000$ (orange) on the noisy ring data in Figure 1 with sample size $10,000$. The imputations by kNNImputer with $k = 5$ are shown as the green dotted vertical line.

mentioned, an imputation of kNNImputer is an estimate of the conditional expectation of a missing response, thus tending to be a deterministic function of the covariate. As a result, the distribution of imputed responses is concentrated around the regression curve, even when the distribution of missing responses has large variability. This is illustrated in Figures 1 and 2, where the true conditional distribution of a missing response is bimodal when the covariate is small, but the distribution of imputations is unimodal and many imputations take values never realized by the true missing values. A substantial bias can occur in an analysis of such a distorted imputed dataset, for example, when estimating the variance, quantiles and modes in the population. (See Sections 2.1 and 2.2 for more formal discussions.)

The above issue of kNNImputer may be addressed by estimating the conditional *distribution* of a missing response given a covariate, and randomly sampling imputations from it. This idea was investigated by Lalande and Doya (2023), who proposed the "kNN×KDE" approach that combines a soft version of kNN and kernel density estimation (KDE). For a given unit, the conditional density of a missing response is estimated as a weighted average of Gaussian densities centered at observed responses, where the weights are computed so that units more similar, in terms of covariates, to the given unit receive larger weights. kNN×KDE was

demonstrated to have good empirical performance in recovering the distribution of missing values, compared to established imputation methods, including kNNImputer, missForest (Stekhoven and Bühlmann, 2012), SoftImpute (Hastie et al., 2015), and Gain (Yoon et al., 2018). However, no theoretical guarantee exists for kNN×KDE, such as its statistical consistency, i.e., whether the estimated conditional density converges to the true one as the sample size increases. Consistency is not only important as a minimal theoretical guarantee but also in understanding how hyperparameters should be chosen. While kNN×KDE has two main hyperparameters (the "inverse temperature" in the softmax function used for weight computations, and the variance of Gaussian densities), no systematic selection procedure was proposed.

This paper studies a simpler kNN-based stochastic imputation method named *kNNSampler*. For a given unit whose response is missing, it estimates the conditional distribution of the missing response given the unit's observed covariate as the *empirical distribution* of the observed responses of the $k$ most similar units to that unit in terms of covariates; an imputation is randomly sampled from this empirical distribution, which we call *kNN conditional distribution*. kNNSampler is as simple as kNNImputer: instead of taking the mean of the observed responses of $k$ nearest neighbours, kNNSampler simply samples one of those $k$ observed responses. It is thus simpler than kNN×KDE as it does not involve an intermediate step of density estimation and is free of any hyperparameter for responses. The number $k$ of nearest neighbours in kNNSampler can be efficiently chosen by leave-one-out cross validation using the fast computation method recently proposed by Kanagawa (2024). Figures 1 and 2 describe imputations by kNNSampler, which align much better with the distribution of true missing values than imputations by kNNImputer. More systematic experiments are provided in Section 4.

kNNSampler can be interpreted as an instance of *hot deck*, classic imputation methods widely used in practice for socio-economic and public health surveys, including the U.S. Census Bureau's Current Population Survey and the National Center for Education Statistics (e.g., Andridge and Little, 2010). In a hot deck method, a missing value of a given unit is imputed as one of the response values of the units belonging to the same "adjustment cell" as the given unit. The method is called *random hot deck* if the imputation is selected randomly from the adjustment cell; it is called *nearest-neighbour hot deck* if nearest neighbours define the adjustment cell (Little and Rubin, 2002, Example 4.9). kNNSampler is thus essentially a nearest-neighbour random hot deck method. However, while classic and widely used, hot deck methods have not been well established theoretically (Andridge and Little, 2010).

Our contribution is to establish kNNSampler, and thus the nearest-neighbour random hot deck, as a theoretically principled missing-value imputation method. To this end, we analyze the kNN conditional distribution, i.e., the empirical distribution of $k$ nearest neighbour responses from which an imputation is sampled, as an estimator of the true conditional distribution of a missing response given a covariate (Section 3). Our theoretical contributions are summarized as follows.

- We derive an error bound between the kNN and true conditional distributions for any given, fixed covariate, in terms of the number $n$ of observed response-covariate pairs, the number $k$ of the nearest neighbours, and other problem-specific constants. The error is measured by the maximum mean discrepancy (MMD) (Gretton et al., 2012), a distance metric on probability distributions that metrizes the weak convergence (Simon-Gabriel et al., 2023), between the kNN and true conditional distributions. It holds under a Lipschitz condition that the response's conditional distribution changes smoothly when the covariate changes continuously. A consequence of the bound is the statistical consistency of the kNN conditional distribution, in that the error decreases to zero as the sample size $n$ goes to infinity, if the number $k$ of nearest neighbours increases to infinity at a rate slower than $n$. This offers a theoretical foundation of the kNNSampler and thus the nearest-neighbour random hot deck.

- To derive the bound, we analyze the mean embedding of the kNN conditional distribution in a reproducing kernel Hilbert space (RKHS) as a novel estimator of the mean embedding of the true conditional distribution, known as *conditional mean embedding* (Muandet et al., 2017, Chapter 4), which is the RKHS-valued regression function (Grünewälder et al., 2012). The RKHS distance between these two embeddings is equivalent to the MMD between the kNN and true conditional

distributions. Our bound leads to the consistency and convergence rates for the novel kNN-based estimator of the conditional mean embedding.

- Our analysis extends the error analysis by Kpotufe (2011) on *real*-valued kNN regression to RKHS-valued regression in which the response variable is infinite-dimensional. As a byproduct, we prove that the required sample size to attain a given level of precision increases exponentially *not* with the covariate's ambient dimension but with the *intrinsic dimension* of the covariate distribution. Therefore, the kNNSampler may not be severely affected by the curse of dimensionality if the covariate distribution has a low intrinsic dimension.

This paper is organised as follows. We describe the proposed approach in Section 2 and its theory in Section 3. We report experimental results on synthetic data in Section 4 and on real solar-power data in Section 5.

## 2 Proposed Approach

This section describes the proposed approach. Section 2.1 introduces the setting. Section 2.2 explains the kNNImputer and its issue as a preliminary. We describe kNNSampler in Section 2.3, uncertainty quantification with the kNN conditional distribution in Section 2.4, and multiple imputation with kNNSampler in Section 2.5.

### 2.1 Setting

We first describe the problem setup. Let $\mathcal{X}$ and $\mathcal{Y}$ be measurable spaces representing the covariate space and the response space, respectively. For example, the covariate space may be the $d$-dimensional Euclidean space, $\mathcal{X} = \mathbb{R}^d$, in which case a covariate $x \in \mathcal{X}$ consists of $d$ features (e.g., a person's age, weight, height), and the response space may be the real line $\mathcal{Y} = \mathbb{R}$, in which case a response $y \in \mathcal{Y}$ is real-valued (e.g., the person's blood pressure).

We assume that our dataset consists of $n + m$ units (e.g., persons), where $n$ units have both covariate $x_i \in \mathcal{X}$ and response $y_i \in \mathcal{Y}$ observed, while $m$ units have only covariate $\tilde{x}_j \in \mathcal{X}$ observed and response $\tilde{y}_{\mathrm{miss},j} \in \mathcal{Y}$ missing:

$$\mathcal{D}_n := \{(x_1, y_1), \ldots, (x_n, y_n)\}, \quad \mathcal{D}_{\mathrm{miss}} := \{(\tilde{x}_1, \tilde{y}_{1,\mathrm{miss}}), \ldots, (\tilde{x}_m, \tilde{y}_{m,\mathrm{miss}})\} \tag{1}$$

For each of the $n$ units with observed responses, we assume that the covariate follows a marginal distribution $P(x)$ and the response given the covariate follows the conditional distribution $P(y|x)$ in an independently and identically distributed (i.i.d.) manner:

$$(x_1, y_1), \ldots, (x_n, y_n) \overset{i.i.d.}{\sim} P(y|x)P(x) \tag{2}$$

On the other hand, for the $m$ units with missing responses, the covariate is assumed to follow a marginal distribution $Q(\tilde{x})$, which can be different from $P(x)$, while the conditional distribution of the missing response given the covariate remains the same:

$$(\tilde{x}_1, \tilde{y}_{1,\mathrm{miss}}), \ldots, (\tilde{x}_m, \tilde{y}_{m,\mathrm{miss}}) \overset{i.i.d.}{\sim} P(\tilde{y}_{\mathrm{miss}}|\tilde{x})Q(\tilde{x}). \tag{3}$$

This assumption implies that the probability of a unit missing its response is determined by the unit's covariate and is not affected by the response. Therefore, it is an instance of the *Missing-At-Random (MAR)* assumption (Rubin, 1976). In the special case where the covariate distributions for the two cases are the same, $Q(\tilde{x}) = P(\tilde{x})$, the assumption can be interpreted as the *Missing-Completely-At-Random (MCAR)* assumption where missingness occurs completely randomly.

Under this setup, missing responses may be imputed by estimating the unknown conditional distribution $P(y|x)$ of a response given a covariate and sampling from it. This is what the kNNSampler does.

**Remark 1.** *Sampling imputations from the conditional distribution is needed for unbiased estimation of a quantity of interest and its well-calibrated uncertainty quantification. We informally describe this from the Bayesian perspective of Rubin (1987), where the quantity of interest, denoted here by θ, is treated as a random variable. For the frequentist perspective, see Rubin (1987; 1996); Murray (2018). A Bayesian analysis is done by computing the posterior distribution of θ given the observed data in (1):*

$$P(\theta \mid \mathcal{D}_n, \ \tilde{x}_1, \dots, \tilde{x}_m)$$
$$= \int \underbrace{P(\theta \mid \mathcal{D}_n, \ (\tilde{x}_1, \tilde{y}_1) \dots, (\tilde{x}_m, \tilde{y}_m))}_{(A)} \cdot \underbrace{P(\tilde{y}_1, \dots, \tilde{y}_m \mid \mathcal{D}_n, \ \tilde{x}_1, \dots, \tilde{x}_m)}_{(B)} \, d\tilde{y}_1 \cdots d\tilde{y}_m,$$

*where (A) is the posterior distribution of θ given observed dataset $\mathcal{D}_n = \{(x_1, y_1), \dots, (x_n, y_n)\}$ and imputed dataset $(\tilde{x}_1, \tilde{y}_1), \dots, (\tilde{x}_m, \tilde{y}_m)$ and is computed by a standard Bayesian analysis, treating the imputations as observed data; (B) is the conditional distribution of missing responses $\tilde{y}_1, \dots, \tilde{y}_m$ given $\mathcal{D}_n$ and observed covariates $\tilde{x}_1, \dots, \tilde{x}_m$ for missing responses, and can be written as*

$$P(\tilde{y}_1, \dots, \tilde{y}_m \mid \mathcal{D}_n, \ \tilde{x}_1, \dots, \tilde{x}_m) = \prod_{i=1}^{m} P(\tilde{y}_i \mid \tilde{x}_i),$$

*where we used (3). Thus, by estimating the conditional distribution $P(\tilde{y} \mid \tilde{x})$ of a response given a covariate and sampling from it, the posterior distribution of θ can be approximately computed, using, e.g., the multiple imputation approach (Rubin, 1987; 1996; Murray, 2018).*

## 2.2 Issue with kNNImputer and Regression-based Imputers

Before describing the proposed kNNSampler, we discuss an issue with the widely used kNNImputer (Troyanskaya et al., 2001) and other regression-based imputation methods.

Suppose that the covariate space $\mathcal{X}$ is equipped with a distance metric $d_{\mathcal{X}}(x, x')$ that quantifies the distance between any two points $x, x' \in \mathcal{X}$. For example, if $\mathcal{X}$ is the Euclidean space, then $d_{\mathcal{X}}(x, x')$ may be the Euclidean distance between two vectors $x$ and $x'$. Let $X_n$ be the set of covariates for the $n$ units with observed responses:

$$X_n := \{x_1, \dots, x_n\}$$

For a given covariate $\tilde{x}$ and a number $k$ of nearest neighbours, let $\mathrm{NN}(\tilde{x}, k, X_n)$ be the indices of the $k$ units whose covariates are the most similar to $\tilde{x}$ in terms of the distance metric among the $n$ units with observed responses:[3]

$$\mathrm{NN}(\tilde{x}, k, X_n) := \{j_1, \dots, j_k \in \{1, \dots, n\} \mid \ d_{\mathcal{X}}(\tilde{x}, x_{j_1}) \leq \dots \leq d_{\mathcal{X}}(\tilde{x}, x_{j_k}) \quad\quad (4)$$
$$\leq d_{\mathcal{X}}(\tilde{x}, x_j) \text{ for all } j \in \{1, \dots, n\} \setminus \{j_1, \dots, j_k\}\}.$$

That is, $\mathrm{NN}(\tilde{x}, k, X_n)$ is the indices of the $k$ nearest neighbours of $\tilde{x}$ in $X_n$.

kNNImputer (Troyanskaya et al., 2001) imputes the missing response $\tilde{y}_{i,\mathrm{miss}}$ of the unit with observed covariate $\tilde{x}_i$ as the average of the observed responses $y_{j_1}, \dots, y_{j_k}$ of its $k$-nearest neighbors $x_{j_1}, \dots, x_{j_k}$:

$$\hat{y}_{i,\mathrm{imp}} = \frac{1}{k} \sum_{j \in \mathrm{NN}(\tilde{x}_i, k, X_n)} y_j.$$

This is kNN regression (e.g., Györfi et al., 2002) and thus estimates the conditional *mean* of the missing response $\tilde{y}_{i,\mathrm{miss}}$ given the observed covariate $\tilde{x}_i$:

$$\hat{y}_{i,\mathrm{imp}} \approx f(\tilde{x}_i) := \int \tilde{y} \, dP(\tilde{y} \mid \tilde{x}_i),$$

---

[3]If there is a tie in the distances $d_{\mathcal{X}}(\tilde{x}, x_i)$, break it randomly.

---

**Algorithm 1:** kNNSampler

---

**Input:** Number of nearest neighbors $k$, observed covariates $\tilde{x}_1, \ldots, \tilde{x}_m \in \mathcal{X}$ with missing responses,
observed covariate-response pairs $(x_1, y_1), \ldots, (x_n, y_n) \in \mathcal{X} \times \mathcal{Y}$.
**Output:** Imputed responses $\hat{y}_{1,\text{imp}}, \ldots, \hat{y}_{m,\text{imp}} \in \mathcal{Y}$.

**for** $i = 1$ **to** $m$ **do**
$\quad \hat{y}_{i,\text{imp}} := y_j$, where $j \in \{1, \ldots, n\}$ is uniformly sampled from $\text{NN}(\tilde{x}_i, k, X_n)$ in equation 4, the indices
$\quad$ of the $k$-nearest neighbors of $\tilde{x}_i$ in $X_n = \{x_1, \ldots, x_n\}$.
**end**

---

where $f : \mathcal{X} \to \mathcal{Y}$ is the regression function. In this case, the observed covariate and the imputed response $(\tilde{x}_i, \hat{y}_{i,\text{imp}})$ approximately follow the degenerate joint distribution

$$\delta(\tilde{y} - f(\tilde{x}))Q(\tilde{x}),$$

where $\delta(\tilde{y} - f(\tilde{x}))$ denotes the Dirac distribution at the conditional mean $f(\tilde{x})$, i.e., the degenerate distribution whose mass concentrates at $f(\tilde{x})$. This differs from the joint distribution of the observed covariate and the true missing response $(\tilde{x}_i, \tilde{y}_{i,\text{miss}})$:

$$P(\tilde{y} \mid \tilde{x})Q(\tilde{x}) \tag{5}$$

unless the conditional distribution $P(\tilde{y} \mid \tilde{x})$ is the Dirac distribution $\delta(\tilde{y} - f(\tilde{x}))$, i.e., unless the missing response is the deterministic function of observed covariate. The same issue occurs with other single imputation methods based on regression, because they impute the missing response by estimating the conditional mean.

To summarize, kNNImputer and other regression-based imputation methods do not generally recover the true distribution of the missing data. An analysis based on the imputed dataset may lead to a biased result. For instance, the variance of the imputed values may be much lower than the variance of the true missing values. kNNSampler alleviates this issue by imputing missing values by estimating the conditional distribution $P(\tilde{y} \mid \tilde{x})$.

**Remark 2.** *Consider the Bayesian analysis in Remark 1. If the missing responses are imputed by deterministic, regression estimates $\hat{y}_{1,\text{imp}}, \ldots, \hat{y}_{m,\text{imp}}$, the posterior distribution becomes that of the quantity of interest $\theta$ given observed and imputed datasets,* **both treated as observed***:*

$$P(\theta \mid \mathcal{D}_n, \ (\tilde{x}_1, \hat{y}_{1,\text{imp}}) \ldots, (\tilde{x}_m, \hat{y}_{m,\text{imp}}))$$

*This ignores uncertainties in the missing responses, leading to final uncertainty estimates for $\theta$ that are overconfident (a prediction interval may be much narrower than the actual error of a point estimate).*

## 2.3 kNNSampler

We now describe kNNSampler (Algorithm 1). Consider imputing the missing response $\tilde{y}_{\text{miss}}$ of a unit with observed covariate $\tilde{x}$. kNNSampler estimates the conditional distribution $P(\tilde{y}_{\text{miss}} \mid \tilde{x})$ of $\tilde{y}_{\text{miss}}$ given $\tilde{x}$ as the empirical distribution of the observed responses $y_{j_1}, \ldots, y_{j_k}$ of the $k$ nearest neighbours $x_{j_1}, \ldots, x_{j_k}$ of $\tilde{x}$:

$$P(\tilde{y}_{\text{miss}} \mid \tilde{x}) \approx \hat{P}(\tilde{y}_{\text{miss}} \mid \tilde{x}) := \frac{1}{k} \sum_{j \in \text{NN}(\tilde{x}, k, X_n)} \delta(\tilde{y}_{\text{miss}} - y_j), \tag{6}$$

which is the discrete distribution where each of $y_{j_1}, \ldots, y_{j_k}$ has probability mass $1/k$. An imputation $\hat{y}_{\text{imp}}$ for the missing response is randomly sampled from this empirical distribution:

$$\hat{y}_{\text{imp}} \sim \hat{P}(\tilde{y}_{\text{miss}} \mid \tilde{x}).$$

Algorithmically, this is to randomly sample one of the kNN observed responses $y_{j_1}, \ldots, y_{j_k}$. Algorithm 1 independently applies this procedure to the observed covariate $\tilde{x}_i$ to generate an imputation $\hat{y}_{i,\text{imp}}$ of missing value $y_{i,\text{miss}}$ for each unit $i = 1, \ldots, m$.

**Choice of $k$**   The number of nearest neighbors $k$ is a hyperparameter of kNNSampler. The theoretical and empirical results below indicate that $k$ should not be fixed to a prespecified value (e.g., $k = 5$), and should be chosen depending on the available data. One way is to perform cross-validation for kNN regression on the data $(x_1, y_1), \ldots, (x_n, y_n)$ and select $k$ among candidate values that minimizes the mean-square error on held-out observed responses, averaged over different training-validation splits. In particular, the present work uses Leave-One-Out Cross-Validation (LOOCV) using the fast computation method recently proposed by Kanagawa (2024).

## 2.4   Uncertainty Quantification of Missing Values

Quantifying the uncertainty in missing values is important for several reasons, including assessing the reliability of imputations and the adequacy of the covariates used, as well as determining how to perform imputations (e.g., single or multiple) and how to use the imputations in subsequent analyses. We describe here how to perform uncertainty quantification of missing values with the kNN conditional distribution.

**Conditional Probability Estimation**   kNNSampler can be used to estimate the conditional probability of a missing response $\tilde{y}_{\text{miss}}$ belonging to a specified (measurable) subset $S$ of the response space $\mathcal{Y}$, given observed covariate $\tilde{x}$:

$$\Pr(\tilde{y}_{\text{miss}} \in S \mid \tilde{x}) = \int \mathbb{I}[\tilde{y} \in S] \, dP(\tilde{y} \mid \tilde{x}),$$

where $\mathbb{I}[\tilde{y} \in S]$ is the indicator function that outputs 1 if $\tilde{y} \in S$ and 0 otherwise. By replacing the unknown conditional distribution $P(\tilde{y} \mid \tilde{x})$ by the kNN conditional distribution $\hat{P}(\tilde{y} \mid \tilde{x})$ in (6), this conditional probability is approximated as

$$\widehat{\Pr}(\tilde{y}_{\text{miss}} \in S \mid \tilde{x}) = \int \mathbb{I}[\tilde{y} \in S] \, d\hat{P}(\tilde{y} \mid \tilde{x}) = \frac{1}{k} \sum_{j \in \text{NN}(\tilde{x}, k, X_n)} \mathbb{I}[y_j \in S].$$

In other words, the conditional probability is estimated as the observed frequency of the kNN response values that fall in $S$.

**Interval Estimation**   Let us focus on a real-valued missing response $\tilde{y}_{\text{miss}} \in \mathcal{Y} = \mathbb{R}$. The conditional probability of the missing response belonging to a given (finite or infinite) interval $S = (\ell, u)$, where $\ell < u$, is estimated as the observed frequency of the k-NN responses belonging to that interval. This indicates that an interval to which the kNN responses belongs at a specified frequency $0 < 1 - \alpha < 1$ (e.g., $\alpha = 0.05$, in which case the 95% of the kNN responses belong to the interval) is an estimate of an interval to which the unknown missing response belongs at that probability $1 - \alpha$.

Such an interval $(\ell, u)$ is constructed by defining its lower bound $\ell$ and upper bound $u$ as, respectively, the lower and upper $\alpha/2$ empirical quantiles of the kNN responses, i.e., the $k\alpha/2$-smallest and the $k\alpha/2$-largest kNN responses (e.g., if $k = 200$ and $\alpha = 0.05$, the 5th smallest and the 5th largest kNN responses):

$$\Pr(\ell < \tilde{y}_{\text{miss}} < u \mid \tilde{x}) \approx 1 - \alpha$$

**Conditional Standard Deviation Estimation**   The conditional standard deviation of a missing response given observed covariate quantifies the variability of the missing response. This can be estimated by the empirical standard deviation of the kNN response values for the observed covariate.

## 2.5   Multiple Imputation with kNNSampler

kNNSampler can be used for multiple imputation by independently generating multiple imputed datasets. More precisely, let $B$ be the number of multiple imputed datasets to be generated (e.g., $B = 10$). For each $b = 1, \ldots, B$, kNNSampler is independently applied to impute the missing responses in the dataset $\mathcal{D}_{\text{miss}}$ (1) to create an imputed dataset

$$\mathcal{D}_{n+m}^{(b)} := \mathcal{D}_n \cup \mathcal{D}_{\text{imp}}^{(b)} \quad \text{where} \quad \mathcal{D}_{\text{imp}}^{(b)} := \{(\tilde{x}_1, \tilde{y}_{1,\text{imp}}^{(b)}), \ldots, (\tilde{x}_m, \tilde{y}_{m,\text{imp}}^{(b)})\},$$

where $\tilde{y}_{i,\text{imp}}^{(b)}$ is an imputation for the $i$-th unit with a missing response $\tilde{y}_{i,\text{miss}}$ covariates $\tilde{x}_i$. This results in $B$ imputed datasets:

$$\mathcal{D}_{n+m}^{(1)}, \ldots, \mathcal{D}_{n+m}^{(B)}.$$

An analysis can then be made based on the standard procedure of multiple imputation (Rubin, 1987).

For example, suppose that we want to estimate a population quantity $\theta^*$ (e.g., the mean customer satisfaction level of a population). Let $S_{n+m}$ be a function of a dataset of size $n+m$ that outputs an estimate $\hat{\theta}_{n+m}$ of the unknown $\theta^*$ (e.g., the empirical average of $n+m$ values): $\hat{\theta}_{n+m} = S(\mathcal{D}_{n+m})$. Apply this function to each of the $B$ imputed datasets, one obtains $B$ estimates of $\theta^*$:

$$\hat{\theta}_{n+m}^{(b)} = S(\mathcal{D}_{n+m}^{(b)}), \quad b = 1, \ldots, B.$$

The empirical average of these $B$ estimates gives a multiple-imputation estimate of $\theta^*$. The empirical standard deviation of the $B$ estimates $\hat{\theta}_{n+m}^{(1)}, \ldots, \hat{\theta}_{n+m}^{(B)}$ quantifies the uncertainty due to the missingness in the original data. Combined with the standard error of each $\hat{\theta}_{n+m}^{(b)}$, this standard deviation can be used to quantify the overall uncertainty of the estimate using Rubin's rule.

## 3 Theory

We describe a theory for kNNSampler's conditional distribution (6) as an estimator of the true conditional distribution. We shall show that, as the number $k$ of nearest neighbors increases at an approximate rate as the increase of the number $n$ of observed covariate-response pairs, the kNN conditional distribution converges to the true one in the Maximum Mean Discrepancy (MMD) (Gretton et al., 2012), which implies the convergence in distribution (Sriperumbudur et al., 2010, Section 5). We prove this by adapting the proof of the convergence rates of real-valued kNN regression by Kpotufe (2011, Theorem 1) to *Hilbert space-valued* kNN regression.[4]

We use the framework of *kernel mean embedding* (Muandet et al., 2017) in which every probability distribution is represented as a distinct point in an infinite-dimensional feature space known as a reproducing kernel Hilbert space (RKHS). The true and kNN conditional distributions are represented as points in an RKHS, and the distance between them, which is the MMD, quantifies the estimation error. An upper bound on this distance is obtained in terms of the sample size, the number of nearest neighbours, and other relevant quantities.

### 3.1 RKHS Embeddings of Conditional Distributions

Let us first define an RKHS on the response space $\mathcal{Y}$. As before, $\mathcal{Y}$ is a measurable space such as the $p$-dimensional Euclidean space, $\mathcal{Y} = \mathbb{R}^p$. A Hilbert space[5] $\mathcal{H}$ consisting of functions $f$ on $\mathcal{Y}$ is called RKHS if there exists a map

$$\Phi : \mathcal{Y} \to \mathcal{H}$$

called *feature map*, such that the value $f(y)$ of any function $f$ in $\mathcal{H}$ at any point $y$ in $\mathcal{Y}$ can be written as the inner product between $f$ and the feature map $\Phi(y)$ of $y$:

$$f \in \mathcal{H} \quad \Longleftrightarrow \quad f(y) = \langle f, \Phi(y) \rangle_{\mathcal{H}} \text{ for all } y \in \mathcal{Y},$$

where $\langle \cdot, \cdot \rangle_{\mathcal{H}}$ denotes the inner product of $\mathcal{H}$. The $\Phi(y)$ may be called *feature vector* of $y$, and $\mathcal{H}$ the *feature space*, which can be infinite-dimensional.

The inner product between the feature maps $\Phi(y), \Phi(y')$ of any two points $y, y'$ defines the *kernel function*

$$\ell(y, y') := \langle \Phi(y), \Phi(y') \rangle_{\mathcal{H}} \quad \text{for all } y, y', \in \mathcal{Y}. \tag{7}$$

---

[4]Hilbert space-valued kNN regression was also analyzed in Lian (2011), but their results are not directly applicable to our case. This is because Lian (2011) assumes that Hilbert space-valued noises are independent of input variables, but this assumption is too strong in our case.

[5]A Hilbert space is a vector space in which an inner product is defined, the norm is induced from the inner product, and the limit point of any convergent sequence in this norm belongs to the vector space.

This is called *reproducing kernel* of the RKHS. The RKHS and the reproducing kernel are one-to-one, so an RKHS can be induced by defining a kernel. For example, if $\mathcal{Y} = \mathbb{R}^p$, the Gaussian kernel $\ell(y, y') = \exp(-\alpha\|y - y'\|^2)$ for $\alpha > 0$ is the reproducing kernel of a certain RKHS $\mathcal{H}$, and there exists an infinite-dimensional feature map $\Phi$ that induces the Gaussian kernel as (7). See e.g. Steinwart and Christmann (2008); Kanagawa et al. (2025) for details on RKHSs.

Every probability distribution $P$ on $\mathcal{Y}$ is represented as the expected feature map:

$$\Phi(P) := \int \Phi(y) dP(y) \in \mathcal{H}.$$

This is called *mean embedding* of $P$. If the RKHS $\mathcal{H}$ is large enough, any two different probability distributions $P$ and $Q$ are mapped to two distinct mean embeddings:

$$P \neq Q \quad \Longleftrightarrow \quad \Phi(P) \neq \Phi(Q).$$

In this case, the RKHS is called *characteristic* (Sriperumbudur et al., 2010). For example, Gaussian, Matérn and Laplace kernels induce characteristics RKHSs.

The true and kNN conditional distributions in (2) and (6) are represented as their mean embeddings:

$$\Phi(P(\cdot \mid x)) := \int \Phi(y) dP(y|x) \quad \text{and} \quad \Phi(\hat{P}(\cdot \mid x)) := \frac{1}{k} \sum_{j \in \text{NN}(x,k,X_n)} \Phi(y_j) \quad \text{for all} \ \ x \in \mathcal{X}. \tag{8}$$

Here, the dot " $\cdot$ " is used in the notation of the conditional distributions to emphasize that they are probability distributions on $\mathcal{Y}$ and do not depend on a specific value of $y \in \mathcal{Y}$. The RKHS distance between the two conditional mean embeddings is the MMD between the true and kNN conditional distributions. It is used as an error metric of the kNN conditional distribution and theoretically analyzed in the following.

The mean embedding of the conditional distribution is known as *conditional mean embedding* (Song et al., 2009; 2013) and its estimator based on a regularized least-squares algorithm has been studied extensively (e.g., Grünewälder et al., 2012; Li et al., 2022; 2024). The mean embedding of the kNN conditional distribution in (8) is a new estimator of the conditional mean embedding. Its analysis below is thus a new contribution to the RKHS literature and may be of independent interest.

## 3.2 Assumptions

We describe key assumptions for the analysis, which follow Kpotufe (2011) with appropriate modifications.

The conditional mean embedding in (8) is the conditional expectation of the response feature vector $\Phi(y)$ given a covariate $x \in \mathcal{X}$; thus, it is the RKHS-valued regression function (Grünewälder et al., 2012). We assume that the map from a covariate $x$ to the conditional mean embedding $\Phi(P(\cdot \mid x))$ is smooth in the sense that it is Lipschitz continuous.

**Assumption 1.** *There exists a constant $\lambda > 0$ such that the RKHS distance between the conditional mean embeddings for any two inputs $x, x' \in \mathcal{X}$ is bounded by the distance between $x$ and $x'$ times $\lambda$:*

$$\|\Phi(P(\cdot \mid x)) - \Phi(P(\cdot \mid x'))\|_{\mathcal{H}} \leq \lambda \, d_{\mathcal{X}}(x, x') \quad \text{for all} \ \ x, x' \in \mathcal{X},$$

*where $\|\cdot\|_{\mathcal{H}}$ is the norm of the RKHS $\mathcal{H}$.*

Our next assumption is that the reproducing kernel (7) is bounded on $\mathcal{Y}$. This is a mild assumption satisfied by many commonly used kernels such as Gaussian, Matérn and Laplace kernels.

**Assumption 2.** *There exists a constant $C_{\text{ker}} > 0$ that upper-bounds the value of the reproducing kernel (7):*

$$0 \leq \ell(y, y') \leq C_{\text{ker}}^2 \quad \text{for all} \ \ y, y' \in \mathcal{Y}.$$

It can be easily shown that this assumption implies that the RKHS distance between the conditional mean embedding and any response's feature vector is bounded:

$$\|\Phi(P(\cdot \mid x)) - \Phi(y)\|_{\mathcal{H}} \leq \sqrt{2} C_{\text{ker}} \quad \text{for all} \ \ x \in \mathcal{X} \text{ and} \ \ y \in \mathcal{Y}. \tag{9}$$

This implies that the "noise" in the RKHS-valued regression is bounded.

The next assumption is about the *intrinsic dimension* of the marginal distribution $P(x)$ on the covariate space, which can be much smaller than the covariate's dimension $p$ if $x \in \mathbb{R}^p$. The error of the kNN conditional distribution shall be shown to decrease as the sample size increases at a rate depending on the intrinsic dimension, not the covariate's dimension. Let $B(x, r) \subset \mathcal{X}$ denote the ball of center $x \in \mathcal{X}$ and radius $r > 0$:

$$B(x, r) := \{x' \in \mathcal{X} \mid d_{\mathcal{X}}(x, x') \leq r\}.$$

**Assumption 3.** *For the marginal distribution $P(x)$ on the covariate space $\mathcal{X}$, there are positive constants $C_{\mathrm{dist}} > 0$, $r_{\max} > 0$, and $d > 0$ such that*

$$P(B(x, r)) \leq C_{\mathrm{dist}} \epsilon^{-d} P(B(x, \epsilon r)) \quad \text{for all } 0 < r < r_{\max} \text{ and all } 0 < \epsilon < 1.$$

This assumption states that if the radius of a ball is increased by a factor of $\epsilon^{-1}$, the probability mass of the ball increases by at most a factor of $(\epsilon^{-1})^d$. Therefore, the constant $d$ is interpreted as the intrinsic dimension of the covariate distribution, and can be much lower than the ambient dimension $p$ if $\mathcal{X} = \mathbb{R}^p$. For example, if the distribution $P(x)$ is supported on a line in a two-dimensional space, then $d = 1$ while $p = 2$. If $P(x)$ is supported on a plane in a three-dimensional space, then $d = 2$ and $p = 3$ and so forth.

Lastly, we need the following technical condition.

**Assumption 4.** *The covariate space $\mathcal{X}$ is a metric space with distance metric $d_{\mathcal{X}}$ such that the class of all balls $\mathcal{B} := \{B(x, r) \mid x \in \mathcal{X}, \ r > 0\}$ has a finite Vapnik–Chervonenkis (VC) dimension $\mathcal{V}_{\mathcal{B}} > 0$.*

This assumption is satisfied, for example, if $\mathcal{X} = \mathbb{R}^p$ with $p \geq 1$, in which case $\mathcal{V}_{\mathcal{B}} \leq p + 2$ (e.g., Mohri et al., 2018, Exercise 3.17).

### 3.3 Error Bounds and Convergence Rates

Under the above assumptions, the distance between the true and kNN conditional distributions can be upper-bounded as follows. The proof, provided in Appendix A, is an adaptation of the proof of Kpotufe (2011, Theorem 1), which is an upper error bound on real-valued kNN regression, to our setting of RKHS-valued regression.

**Theorem 1.** *Let $(x_1, y_1), \ldots, (x_n, y_n) \overset{i.i.d.}{\sim} P(y|x)P(x)$ and $\hat{P}(y|x)$ be the kNN conditional distribution (6) with $k$ nearest neighbours. Suppose that Assumptions 1, 2, 3 and 4 hold. Let $0 < \delta < 1$. Then, with probability at least $1 - 2\delta$, the bound*

$$\left\| \Phi(P(\cdot \mid x)) - \Phi(\hat{P}(\cdot \mid x)) \right\|_{\mathcal{H}}^2 \leq 4C_{\mathrm{ker}}^2 (1 + 4(\mathcal{V}_{\mathcal{B}} \ln(n) - \ln(\delta)) \cdot \frac{1}{k} + 2\lambda^2 r^2 \left( \frac{3C_{\mathrm{dist}}}{P(B(x, r))} \cdot \frac{k}{n} \right)^{2/d} \tag{10}$$

*holds simultaneously for all $x \in \mathcal{X}$, $k \in \{1, \ldots, n\}$ and $0 < r < r_{\max}$ satisfying*

$$k \geq \mathcal{V}_{\mathcal{B}} \ln(2n) + \ln(8/\delta) \quad \text{and} \quad \frac{k}{n} < \frac{P(B(x, r))}{3C_{\mathrm{dist}}}. \tag{11}$$

From Theorem 1, the following observations can be made.

**Consistency.** Focusing on the dependence on the sample size $n$ and the number $k$ of nearest neighbours, the bound (10) can be written as

$$\left\| \Phi(P(\cdot \mid x)) - \Phi(\hat{P}(\cdot \mid x)) \right\|_{\mathcal{H}}^2 \leq C_1 \frac{\ln(n)}{k} + C_2 \left( \frac{k}{n} \right)^{2/d}, \tag{12}$$

where $C_1$ and $C_2$ are constants independent of $n$ and $k$. The first and second terms correspond to the variance and bias, respectively, of the kNN-based conditional mean embedding estimator $\Phi(\hat{P}(\cdot \mid x))$. The overall error decreases to zero as $n$ increases if both the variance and bias decrease to zero; this requires that

$k$ increases as $n$ increases so that the variance goes to zero, $\ln(n)/k \to 0$, while $k$ should not decrease "too fast" so that the bias also goes to zero, $k/n \to 0$:

$$\left\| \Phi(P(\cdot \mid x)) - \Phi(\hat{P}(\cdot \mid x)) \right\|_{\mathcal{H}} \longrightarrow 0 \quad \text{as} \quad n \to \infty \quad (\text{with } k/n \to 0 \text{ and } \ln(n)/k \to 0) . \tag{13}$$

On the other hand, if $k$ is fixed to a constant value (e.g., $k = 1$), the variance term does not decrease even if the sample size increases. These observations are well known for real-valued kNN regression (e.g., Györfi et al., 2002).

**Convergence in Distribution.** The above consistency (13) implies the *convergence in distribution* (or *weak convergence*) of the kNN conditional distribution $\hat{P}(\cdot \mid x)$ to the true one $P(\cdot \mid x)$ if the response space $\mathcal{Y}$ is a compact metric space (e.g., $\mathcal{Y}$ is a bounded closed subset of an Euclidean space) and $\mathcal{H}$ is a universal RKHS[6], such as the RKHSs of Gaussian, Matérn and Laplace kernels (Sriperumbudur et al., 2010, Theorem 23); see Simon-Gabriel et al. (2023) for more generic conditions. That is, under these conditions, the expectation of any continuous bounded function $f : \mathcal{Y} \to \mathbb{R}$ under the kNN distribution $\hat{P}(\cdot \mid x)$ converges to the expectation under the true distribution $P(\cdot \mid x)$:

$$\int f(y) d\hat{P}(y \mid x) \longrightarrow \int f(y) dP(y \mid x) \quad \text{as} \quad n \to \infty \quad (\text{with } k/n \to 0 \text{ and } \ln(n)/k \to 0) .$$

This supports using the approximate conditional distribution in multiple imputation of missing values.

**Convergence Rates.** An asymptotically optimal choice of $k$ that minimizes the bound (12), up to the $\ln(n)$ factor, can be obtained by balancing the variance and bias terms. If we set $k \propto n^{\frac{2}{2+d}}$, we obtain the convergence rate

$$\left\| \Phi(P(\cdot \mid x)) - \Phi(\hat{P}(\cdot \mid x)) \right\|_{\mathcal{H}}^2 \leq C_3 \ln(n) \cdot n^{-\frac{2}{2+d}}, \tag{14}$$

where $C_3$ is a constant independent of $n$ and $k$.

The rate (14) shows that the required sample size $n$ to attain a desired error level increases exponentially with respect to the intrinsic dimension $d$ of the covariate distribution $P(x)$, not the ambient dimension of the input space $\mathcal{X}$, which is captured by the VC dimension $\mathcal{V}_\mathcal{B}$ of all the balls in $\mathcal{X}$. Therefore, even when the covariate's dimension is large, the error can be small if the covariate features have strong correlations so that the intrinsic dimension $d$ is small. This is the finding first made by Kpotufe (2011) on real-valued kNN regression, and we extend it to RKHS-valued kNN regression.

The rate (14) is the same as the minimax optimal rate for estimating a Lipschitz-continuous *real*-valued regression function when the covariate distribution $P(x)$ has the intrinsic dimension $d$ (Kpotufe, 2011, Theorem 2). An interesting point is that the same rate is attained with RKHS-valued kNN regression where the output space is an RKHS that can be infinite-dimensional. Similar observations have been made for RKHS-valued kernel ridge regression (Li et al., 2022; 2024).

**Implication to Missing Value Imputation.** The second inequality in the condition (11) implies that, for successful recovery of the missing value distribution, the support of the covariate distribution $Q(x)$ for units with missing responses (see (3)) should be reasonably covered by the support of the covariate distribution $P(x)$ for units with observed responses. To explain this, suppose that a missing-response unit has covariate $x'$, i.e., $x'$ is in the support of $Q(x)$, but $x'$ is not in the support of $P(x)$ so that there exists some $r' > 0$ with $P(B(x', r')) = 0$; then the condition (11) is not satisfied for any $n$ and $k$.

## 4 Synthetic Data Experiments

We describe experiments to assess the empirical performance of kNNSampler in recovering the distribution of missing values. Section 4.1 explains the settings, evaluation metrics, and benchmark methods. Section 4.2 describes and discusses the results.

---

[6]An RKHS $\mathcal{H}$ consisting of functions on a metric set $\mathcal{Y}$ is called *universal* if any continuous bounded function $f : \mathcal{Y} \to \mathbb{R}$ can be approximated arbitrarily well in terms of the supremum norm by functions in $\mathcal{H}$.

### 4.1 Settings, Evaluation Metrics and Benchmarks

#### 4.1.1 Data Settings

We consider the following two models for data generation. As before, let $n$ be the number of units with observed responses, $m$ be the number of units with missing responses, and $N = n + m$ be the total number of units.

**Setup 1 (Linear with Chi-square noise).** For each unit $i = 1, \ldots, N$, covariate $x_i$ is uniformly randomly generated on the interval $[-2, 2]$. Response $y_i$ is the sum of covariate $x_i$ and noise $\epsilon_i$ generated randomly from the chi-square distribution with degree of freedom 2:

$$y_i = x_i + \epsilon_i, \quad \text{where} \quad x_i \sim \text{unif}([-2, 2]), \quad \epsilon_i \sim \chi^2(2). \tag{15}$$

Since chi-square noises are positive, this setup enables assessing the capability of imputation methods to recover non-Gaussian, asymmetric data distributions.

**Setup 2 (Noisy 2D ring).** This model, considered by Lalande and Doya (2023), randomly generates covariate $x_i$ and response $y_i$ for each unit $i = 1, \ldots, N$ from a noisy two-dimensional ring of unit radius perturbed with an additive Gaussian noise of variance 0.1:

$$y_i = (1 + \epsilon_i)\sin(\theta_i), \quad x_i = (1 + \epsilon_i)\cos(\theta_i), \quad \text{where} \quad \theta_i \sim \text{unif}[0, 2\pi], \quad \epsilon_i \sim \mathcal{N}(0, 0.1). \tag{16}$$

The conditional distribution of response $y_i$ given covariate $x_i$ is bi-modal when $x_i$ is between about $-0.5$ and $0.5$. Thus, this setup enables the assessment of imputation methods in recovering a multi-modal missing-value distribution.

**Missing Data Mechanism** We consider the MAR (missing at random) setting.[7] We select $m$ units uniformly randomly from the subset of the $N$ units whose covariates lie on the interval $[0.5, 1.5]$ and make their responses missing. We set $m = 200$, and vary $n$ to assess the effect of training size on imputation performance. Specifically, we set $n \in \{2800, 4800, 6800, 8800, 10800\}$.

#### 4.1.2 Performance Metric: Energy Distance

To quantify the performance of an imputation method in recovering the missing value distribution, we compute the energy distance (Székely and Rizzo, 2013) between the empirical distributions of the complete and imputed datasets. We use the energy distance as it is a proper distance between distributions, can be easily computed from samples based on their Euclidean distances without the need for optimization (as compared with, e.g., a Wasserstein distance whose computation requires optimization to solve optimal transport (Peyré et al., 2019)), and is parameter-free (in contrast to the MMD defined with, e.g., a Gaussian kernel, which depends on the bandwidth parameter). The energy distance is a canonical instance of MMD defined with a distance-based kernel (Sejdinovic et al., 2013): it is "canonical" in the sense that it is both scale-invariant (the distance scales linearly with the scale of data) and rotation-invariant (Székely and Rizzo, 2013, Section 3).

Let $\tilde{x}_1, \ldots, \tilde{x}_m$ be the covariates of the $m$ units whose responses $\tilde{y}_1, \ldots, \tilde{y}_m$ are missing, and $\tilde{y}_1^*, \ldots, \tilde{y}_m^*$ be their imputations. For each unit $i$, let $z_i = (\tilde{x}_i, \tilde{y}_i)$ be the pair of the covariate and the true (missing) response, and $z_i^* = (\tilde{x}_i, \tilde{y}_i^*)$ be the pair of the covariate and the imputation. We compute the energy distance between the empirical distributions of $D_m := \{z_1, \ldots, z_m\}$ and $D_m^* := \{z_1^*, \ldots, z_m^*\}$ as

$$\mathcal{E}(D_m, D_m^*) := \frac{2}{m^2} \sum_{i,j=1}^{m} \|z_i - z_j^*\| - \frac{1}{m(m-1)} \sum_{i \neq j} \|z_i - z_j\| - \frac{1}{n(n-1)} \sum_{i \neq j} \|z_i^* - z_j^*\|.$$

This is an unbiased estimate of the squared energy distance between the two joint distributions $Q(x, y) = P(y|x)Q(x)$ and $Q^*(x, y) = P^*(y|x)Q(x)$, where $P(y|x)$ is the true conditional distribution of true response

---

[7]We also performed the experiments under the MCAR (missing completely at random) setting, but the results were similar and thus omitted.

$y$ given covariate $x$, $P^*(y|x)$ is the conditional distribution of imputed response $y$ given covariate $x$, and $Q(x)$ is the covariate distribution of missing units:

$$\mathcal{E}(Q, Q^*) := 2\mathbb{E}\|z - z^*\| - \mathbb{E}\|z - z'\| - \mathbb{E}\|z^* - z^{*'}\|,$$

where $z, z' \overset{i.i.d.}{\sim} Q$ and $z^*, z^{*'} \overset{i.i.d.}{\sim} Q^*$.

A lower energy distance indicates that the two joint distributions are more similar, implying better recovery of the missing-value distribution. A higher energy distance indicates that the imputed distribution is more dissimilar to the true data distribution.

### 4.1.3 Benchmark Imputation Methods

We compare kNNSampler with the following kNN-based and other imputation methods.

**Linear Imputation:** This method models the response-covariates relation as linear and imputes a missing response by its linear prediction applied to an observed covariate. It should be regarded as a benchmark slightly more sophisticated than naive methods such as mean imputation.

**Random Forest** (Stekhoven and Bühlmann, 2012): This method, widely used in practice, imputes a missing response by averaging its multiple predictions made by bootstrap-sampled tree regressors. It can learn a nonlinear relation between the response and covariate and handle the interactions among covariate features (e.g., Shah et al., 2014; Tang and Ishwaran, 2017). We use the default configuration in `scikit-learn`.

**kNNImputer** (Troyanskaya et al., 2001): See Section 2.2 for the description of the method. We set the number $k$ of nearest neighbours as $k = 5$, which is the default setting in `scikit-learn` and widely used in practice.

**kNN×KDE (Lalande and Doya, 2023)**: As explained earlier, this method generates an imputation by sampling from an estimated conditional density of a missing response given a covariate. The conditional density is estimated by weighted Gaussian kernel density estimation over observed responses, with weights derived from a softmax function applied to covariate distances. We use the authors' recommended settings: inverse temperature $\tau = 50$ and kernel bandwidth $h = 0.03$.

As suggested earlier, the number $k$ of nearest neighbours for kNNSampler is determined by the fast leave-one-out cross-validation method of Kanagawa (2024) using the observed covariate-response pairs.

## 4.2 Results

### 4.2.1 Qualitative Comparisons

Figures 3 and 4 describe imputation results by the different methods on datasets generated from the linear chi-square model (15) and the noisy ring model (16), respectively, with sample size $N = 10,000$ and 30% missing rate under the MAR mechanism. The results under the MCAR mechanism are similar and omitted.

The linear imputations ignore the variability in the missing responses and demonstrate the danger of naive imputation methods, such as mean and zero imputations. The imputations by Random Forest and kNNImputer appear to be better than the linear imputations, but are distributed more narrowly than the distribution of missing responses. This is evident for the noisy ring dataset (Figure 4), for which the imputed responses lie inside the ring, which is outside the support of the missing value distribution. This happens because these imputation methods estimate the conditional mean of the missing response given a covariate.

kNNSampler and kNN×KDE recover the distribution of missing values much better than the above imputation methods. However, kNN×KDE generated imputations for the linear chi-square model (Figure 3) outside the support of the missing value distribution. This is because the noises in this dataset are asymmetric and non-Gaussian, while kNN×KDE uses Gaussian noises for generating imputations. In contrast, kNNSampler appears to recover the missing-value distributions accurately. We will next quantitatively compare these methods.

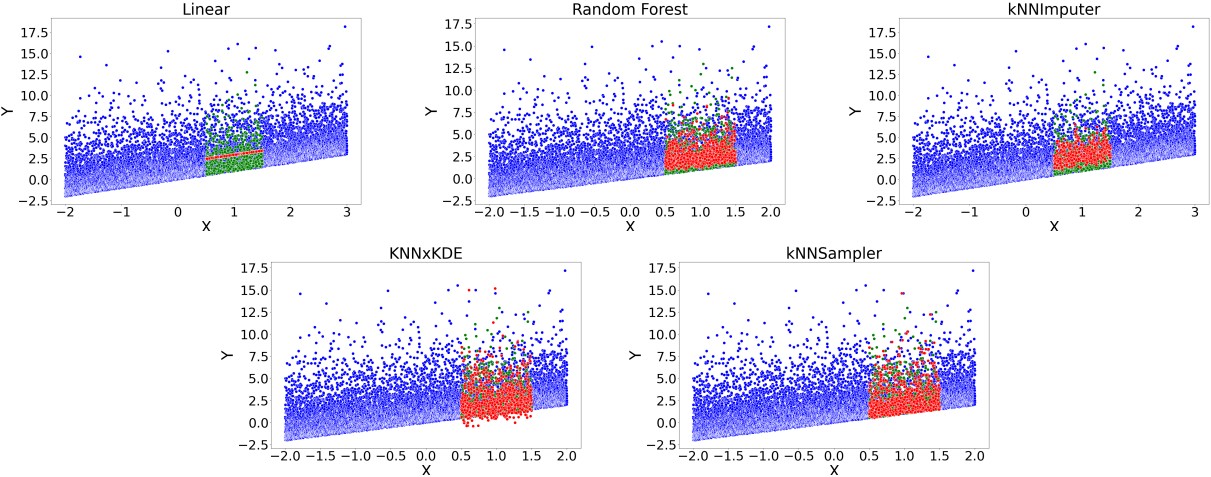

Figure 3: Missing value imputations by different methods for a dataset from the linear chi-square model (15) with sample size $N = 10,000$ with 30% missing rate under the MAR mechanism. True missing responses are shown in green, imputations in red, and the rest in blue.

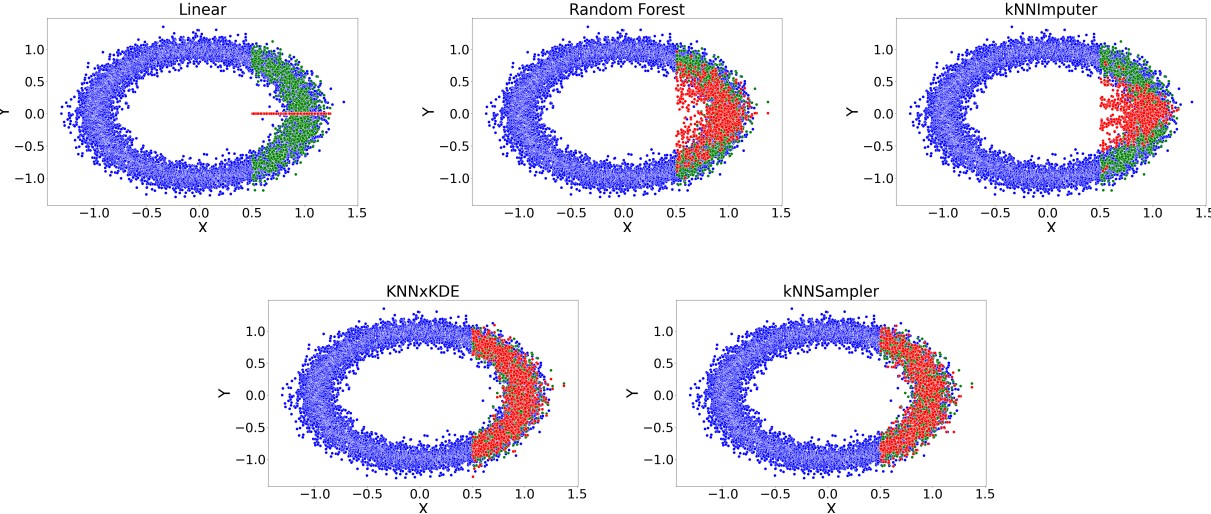

Figure 4: Missing value imputations by different methods for a dataset from the noisy ring model (16) with sample size $N = 10,000$ with 30% missing rate under the MAR mechanism. True missing responses are shown in green, imputations in red, and the rest in blue.

### 4.2.2 Quantitative Comparisons

Each experiment, consisting of data generation, imputations by each method, and the calculation of the evaluation metric, was independently repeated 10 times, and the mean and standard deviation of the evaluation metric are reported. Tables 1 and 2 report the results on the energy distance between the empirical distributions of the imputed and true missing values. See Section 4.1.2 for details.

kNNSampler and kNN×KDE yielded significantly smaller energy distances than the other methods, which suggests that their imputations are distributed more similarly with the true missing values and align with Figures 3 and 4. The energy distance for the linear imputer is the highest among the different methods, quantifying the large discrepancy between the distributions of the imputations and true missing values, as visually observed in Figures 3 and 4. The energy distances for kNNImputer and Random Forest are lower

Table 1: The energy distance between the empirical distributions of imputations and true missing values on the linear chi-square dataset (15). For each method and sample size, the average and standard deviation over 10 independent runs are shown.

| Sample Size | kNNSampler | Random Forest | kNNImputer | kNN×KDE | Linear |
|---|---|---|---|---|---|
| 3000 | **0.027 ± 0.031** | 0.076 ± 0.023 | 0.200 ± 0.038 | **0.036 ± 0.033** | 0.585 ± 0.053 |
| 5000 | **0.027 ± 0.009** | 0.077 ± 0.030 | 0.199 ± 0.041 | **0.033 ± 0.019** | 0.598 ± 0.025 |
| 7000 | **0.027 ± 0.021** | 0.080 ± 0.018 | 0.219 ± 0.024 | **0.028 ± 0.018** | 0.589 ± 0.034 |
| 9000 | **0.017 ± 0.009** | 0.076 ± 0.023 | 0.183 ± 0.031 | **0.016 ± 0.007** | 0.605 ± 0.054 |
| 11000 | **0.018 ± 0.011** | 0.080 ± 0.033 | 0.198 ± 0.040 | **0.026 ± 0.021** | 0.584 ± 0.034 |

Table 2: The energy distance between the empirical distributions of imputations and true missing values on the noisy ring dataset (16). For each method and sample size, the average and standard deviation over 10 independent runs are shown.

| Sample Size | kNNSampler | Random Forest | kNNImputer | kNN×KDE | Linear |
|---|---|---|---|---|---|
| 3000 | **0.021 ± 0.015** | 0.076 ± 0.025 | 0.181 ± 0.032 | **0.033 ± 0.017** | 0.584 ± 0.038 |
| 5000 | **0.019 ± 0.015** | 0.069 ± 0.023 | 0.216 ± 0.055 | **0.024 ± 0.013** | 0.576 ± 0.042 |
| 7000 | **0.028 ± 0.009** | 0.087 ± 0.031 | 0.189 ± 0.032 | **0.028 ± 0.015** | 0.612 ± 0.044 |
| 9000 | **0.028 ± 0.022** | 0.074 ± 0.027 | 0.197 ± 0.043 | **0.020 ± 0.013** | 0.593 ± 0.033 |
| 11000 | **0.019 ± 0.012** | 0.075 ± 0.027 | 0.194 ± 0.064 | 0.035 ± 0.040 | 0.606 ± 0.062 |

Table 3: The root mean squared error of each imputation method for different sample sizes on the linear chi-square dataset (15). The mean and standard deviation over 10 independent runs are shown for each setting.

| Sample Size | kNNSampler | Random Forest | kNNImputer | kNN×KDE | Linear |
|---|---|---|---|---|---|
| 3000 | 2.691 ± 0.151 | 2.338 ± 0.154 | **2.117 ± 0.158** | 2.876 ± 0.126 | **1.885 ± 0.195** |
| 5000 | 2.710 ± 0.134 | 2.273 ± 0.113 | **2.092 ± 0.123** | 2.726 ± 0.190 | **1.914 ± 0.088** |
| 7000 | 2.729 ± 0.102 | 2.307 ± 0.118 | **2.100 ± 0.185** | 2.789 ± 0.135 | **1.895 ± 0.121** |
| 9000 | 2.786 ± 0.228 | 2.308 ± 0.076 | **2.065 ± 0.097** | 2.812 ± 0.095 | **1.945 ± 0.076** |
| 11000 | 2.793 ± 0.188 | 2.388 ± 0.127 | **2.055 ± 0.116** | 2.708 ± 0.184 | **1.913 ± 0.154** |

Table 4: The root mean squared error of each imputation method for different sample sizes on the noisy ring dataset (16). The mean and standard deviation over 10 independent runs are shown for each setting.

| Sample Size | kNNSampler | Random Forest | kNNImputer | kNN×KDE | Linear |
|---|---|---|---|---|---|
| 3000 | 2.680 ± 0.238 | 2.309 ± 0.133 | **2.073 ± 0.169** | 2.811 ± 0.112 | **1.951 ± 0.108** |
| 5000 | 2.818 ± 0.195 | 2.322 ± 0.141 | **2.079 ± 0.157** | 2.698 ± 0.130 | **1.870 ± 0.123** |
| 7000 | 2.733 ± 0.216 | 2.307 ± 0.141 | **2.133 ± 0.163** | 2.799 ± 0.186 | **1.959 ± 0.179** |
| 9000 | 2.638 ± 0.146 | 2.281 ± 0.103 | **2.138 ± 0.141** | 2.637 ± 0.177 | **1.923 ± 0.157** |
| 11000 | 2.672 ± 0.137 | 2.281 ± 0.152 | **2.024 ± 0.075** | 2.763 ± 0.164 | **1.885 ± 0.152** |

than those of the linear imputer, but they are still significantly higher than those of the two other methods. This is reasonable because they are estimating the conditional mean of the missing response given a covariate.

For comparison, we also report the root mean squared error (RMSE) for each method's imputations in Tables 3 and 4. RMSE is expected to be smaller for regression-based methods, which estimate the conditional means of missing values and thereby minimize RMSE. A smaller RMSE does not imply better recovery of the missing-value distribution. Indeed, imputations from the linear imputer have the lowest RMSEs, but their distribution significantly differs from the distribution of true missing values, as quantified in Figures 1 and 2 and visually observed in Figures 3 and 4. This result demonstrates that the RMSE is not a good metric for evaluating the distributional similarity between imputations and missing values. See Näf et al. (2023) for a related discussion.

### 4.3 kNNSampler Uncertainty Quantification

This section evaluates kNNSampler's ability to quantify uncertainty in missing values, using the approach described in Section 2.4. Figure 5 shows the mean and standard deviation of the coverage probabilities of kNN prediction intervals over 10 independent runs, for each sample size and missing rate (MR). As the sample size increases, the coverage probabilities converge to the designed probabilities (80%, 90%, 95%) irrespective of the missing rate, supporting the validity of the prediction intervals.

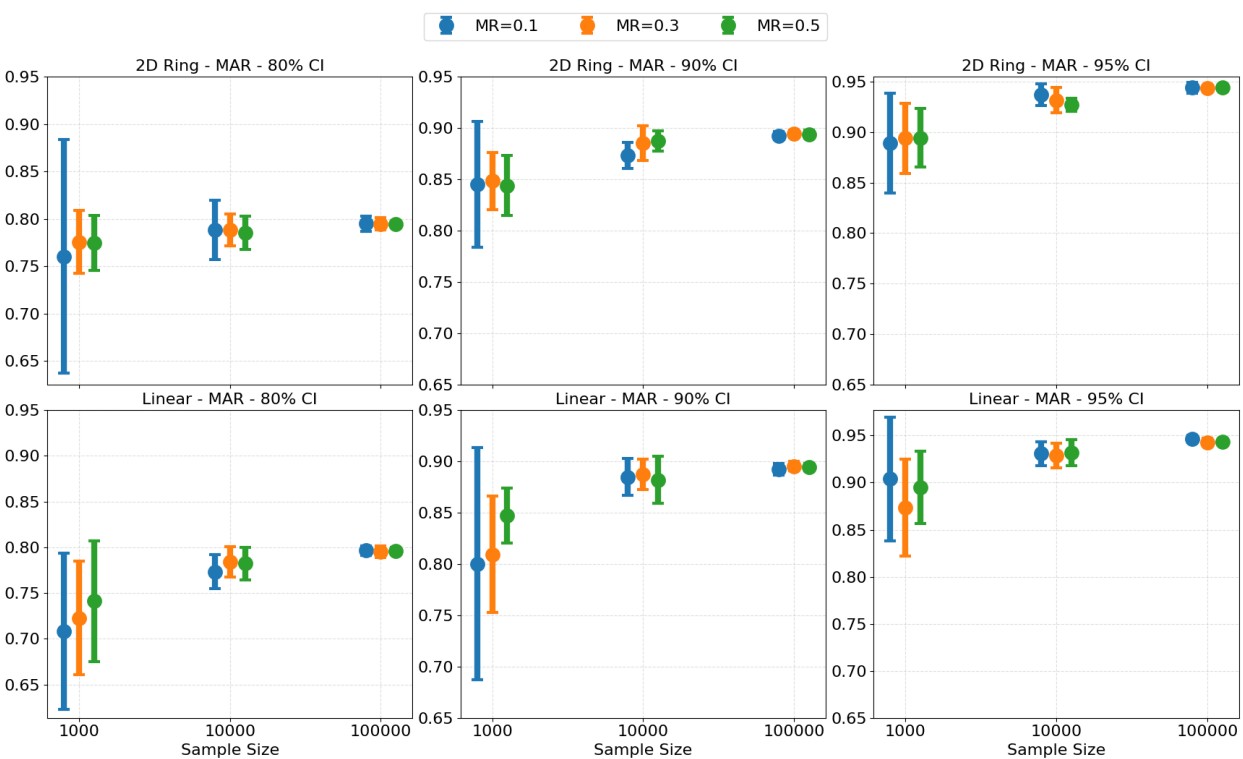

Figure 5: Coverage probabilities of kNN prediction intervals at different missing rates (MR) for different sample sizes. The mean and standard deviation over 10 independent runs are shown for each setting. The top three figures are on the noisy ring data, and the bottom three are on the linear chi-square data.

## 5 Real Data Experiments

Lastly, we present real-data experiments on solar power generated by photovoltaic panels, where missing values are common due to sensor failures and other factors (e.g., Phan et al., 2023; Costa et al., 2024). We use a Kaggle dataset[8] that contains solar panel DC powers (responses) and the corresponding irradiations (covariates), totaling 67,698 covariate-response pairs. We randomly select a subset of $N$ covariate-response pairs from the full dataset. In this subset, we select randomly 30% of the units whose covariates are between 0.4 and 0.6 and set their responses to missing. These missing responses are imputed based on the remaining observed covariate-response pairs in the subset. We consider each of $N \in \{10,000, 20,000, 30,000, 40,000, 50,000, 60,000\}$. The configuration of each method follows Section 4.1.3.

This experiment is repeated 10 times independently for each setting, and the mean and standard deviation of the energy distance between imputations and true missing values are reported in Table 5 (see Section 4.1.2). kNNSampler consistently gives lower energy distances than the other methods, this time in-

---

[8]https://www.kaggle.com/datasets/samuelkamau/solar-data/

cluding kNN×KDE. Moreover, kNNSampler's energy distance decreases as the sample size increases, which aligns with its theoretical consistency in recovering missing-value distributions.

To understand the results, Figure 6 describes imputations by kNNSampler, kNN×KDE, and kNNImputer based on the full dataset. kNN×KDE's imputations do not capture well the heterogeneity and non-negativity of the missing-value distribution, as the imputations are sampled from Gaussian distributions with a fixed, common variance. kNNImputer's imputations are distributed more narrowly than the missing-value distribution. In contrast, kNNSampler's imputations are distributed similarly to the true missing values, successfully recovering the missing-value distribution.

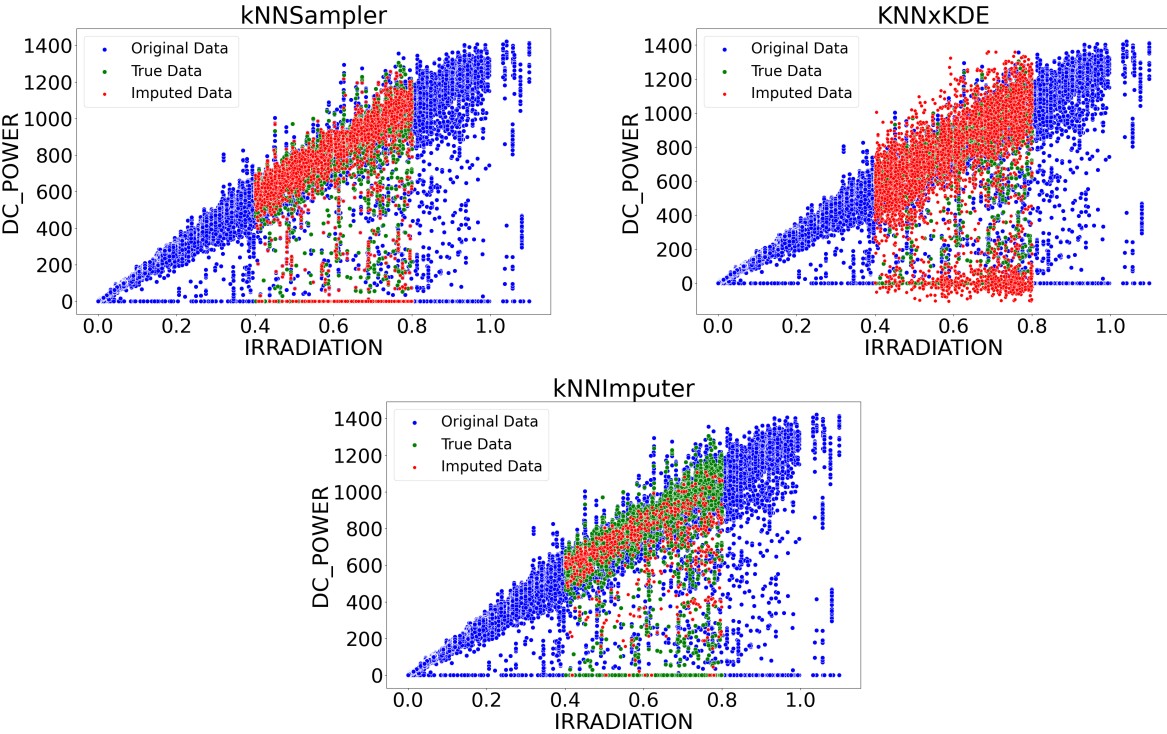

Figure 6: Missing value imputations by kNNSampler, kNN×KDE, and kNNImputer on the full solar panel dataset in Section 5 with 30% missing rate under the MAR mechanism. True missing responses are shown in green, imputations in red, and the rest in blue.

Table 5: Comparison of the energy distance between the empirical distributions of imputations and true missing values across different sample sizes of the real solar panel dataset in Section 5. For each method and sample size, the average and standard deviation of the energy distance over 10 independent runs are shown.

| Sample Size | kNNSampler | Random Forest | kNNImputer | kNN×KDE | Linear |
|---|---|---|---|---|---|
| 10000 | $\mathbf{1.855 \pm 1.474}$ | $5.544 \pm 1.916$ | $10.619 \pm 2.958$ | $3.333 \pm 2.609$ | $190.546 \pm 16.701$ |
| 20000 | $\mathbf{0.687 \pm 0.671}$ | $4.980 \pm 0.977$ | $7.389 \pm 1.763$ | $2.634 \pm 1.238$ | $195.623 \pm 9.364$ |
| 30000 | $\mathbf{0.500 \pm 0.309}$ | $5.112 \pm 1.207$ | $4.874 \pm 1.144$ | $2.273 \pm 0.990$ | $195.412 \pm 3.468$ |
| 40000 | $\mathbf{0.373 \pm 0.317}$ | $5.543 \pm 0.684$ | $4.584 \pm 0.890$ | $2.293 \pm 1.165$ | $194.081 \pm 5.721$ |
| 50000 | $\mathbf{0.190 \pm 0.138}$ | $5.667 \pm 0.968$ | $4.161 \pm 0.881$ | $2.559 \pm 1.133$ | $198.261 \pm 4.039$ |
| 60000 | $\mathbf{0.148 \pm 0.077}$ | $6.230 \pm 0.812$ | $4.634 \pm 1.053$ | $1.927 \pm 0.942$ | $194.808 \pm 3.526$ |

## 6 Conclusion and Discussion

We studied kNNSampler, a stochastic missing-value imputation method that imputes a missing response of a given unit by searching for its $k$ most similar units in terms of covariates and by randomly sampling one

of the associated $k$ observed responses. This method is interpreted as sampling from an approximate kNN-based conditional distribution of a missing response given a covariate. Assuming a Lipschitz condition that the true conditional distribution changes continuously with covariates, we proved that the kNN conditional distribution converges to the true conditional distribution as the number $k$ of nearest neighbours increases at a rate slower than the sample size increases. This analysis offers a theoretical justification for kNNSampler, and may be of independent as it analyzes a novel kNN-based estimator of the Hilbert space embedding of a conditional distribution. Empirical results demonstrate the capability of kNNSampler in recovering the distributions of missing values.

We discuss limitations of the current work, some of which stem from hot-deck methods in general (see Andridge and Little 2010), and potential future directions. (i) While the experiments show the promising performance of kNNSampler, they are limited to low-dimensional covariates. Experiments with higher-dimensional covariates are needed to fully characterize the KNNSampler's practical performance. (ii) For higher-dimensional covariates, the choice of the distance function itself influences kNNSampler's performance, which should be investigated. (iii) Leave-one-out cross-validation for selecting the number of nearest neighbours uses the mean square error, which should be modified to a distributional error metric. (iv) A further theoretical analysis is needed to understand how the distributional imputation quality affects subsequent analysis of a quantity of interest, such as the well-calibratedness of uncertainty estimates.

### Acknowledgments

We thank Frédéric Coutellier, Michele Bezzi and colleagues at SAP Labs France for their support and discussion. We also thank the reviewers and the Action Editor for their time and constructive feedback.

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

## A  Proof of Theorem 1

*Proof.* We proceed as the proof of Kpotufe (2011, Theorem 1) on real-valued kNN regression, with adaptations to our RKHS-valued kNN regression setting.

The RKHS distance between the mean embeddings of the true and kNN conditional distributions is decomposed into the "bias" and "variance" terms:

$$\left\| \Phi(P(\cdot \mid x)) - \Phi(\hat{P}(\cdot \mid x)) \right\|_{\mathcal{H}}^2$$

$$= \left\| \Phi(P(\cdot \mid x)) - \mathbb{E}[\Phi(\hat{P}(\cdot \mid x)) \mid X_n] \ + \ \mathbb{E}[\Phi(\hat{P}(\cdot \mid x)) \mid X_n] - \Phi(\hat{P}(\cdot \mid x)) \right\|_{\mathcal{H}}^2,$$

$$\leq 2 \underbrace{\left\| \Phi(P(\cdot \mid x)) - \mathbb{E}[\Phi(\hat{P}(\cdot \mid x)) \mid X_n] \right\|_{\mathcal{H}}^2}_{\text{Bias}} \ + \ 2 \underbrace{\left\| \mathbb{E}[\Phi(\hat{P}(\cdot \mid x)) \mid X_n] - \Phi(\hat{P}(\cdot \mid x)) \right\|_{\mathcal{H}}^2}_{\text{Variance}}, \tag{17}$$

where $\mathbb{E}[\Phi(\hat{P}(\cdot \mid x)) \mid X_n]$ is the conditional expectation of $\Phi(\hat{P}(\cdot \mid x))$ given $X_n = (x_1, \ldots, x_n)$, the expectation being taken for the $n$ output values $y_1, \ldots, y_n$:

$$\mathbb{E}[\Phi(\hat{P}(\cdot \mid x)) \mid X_n] = \frac{1}{k} \sum_{j \in \mathrm{NN}(x,k,X_n)} \mathbb{E}[\Phi(y_j) \mid X_n] = \frac{1}{k} \sum_{j \in \mathrm{NN}(x,k,X_n)} \Phi(P(\cdot \mid x_j)), \tag{18}$$

where the last identity follows from $y_j \sim P(\cdot \mid x_j)$.

Lemma 2 in Section A.1 and Lemma 3 in Section A.2 respectively provide probabilistic upper bounds of the bias and variance terms in the upper bound (17), each holding simultaneously for all $x \in \mathcal{X}$, $k \in \{1, \ldots, n\}$ and $r > 0$ satisfying the condition (11) with probability at least $1 - \delta$. The claim follows from using these probabilistic bounds in (17).

$\square$

## A.1 Bias Bound

Lemma 1 below is from Kpotufe (2011, Lemma 1).

**Lemma 1.** *Suppose that Assumption 4 holds. Let $x_1, \ldots, x_n \overset{i.i.d.}{\sim} P(x)$ be an i.i.d. sample of size $n$ from a probability distribution $P$ on $\mathcal{X}$, and $P_n := \frac{1}{n} \sum_{i=1}^n \delta_{x_i}$ be the empirical distribution. Let $0 < \delta < 1$. Then,*

$$P_n(B) = \frac{1}{n} \sum_{i=1}^n \mathbb{I}[x_i \in B] \geq a$$

*holds simultaneously for all balls $B \in \mathcal{B}$ and for all constants $a > 0$ satisfying*

$$P(B) \geq 3a \quad and \quad a \geq \frac{\mathcal{V}_{\mathcal{B}} \ln(2n) + \ln(8/\delta)}{n}.$$

*with probability at least $1 - \delta$.*

**Lemma 2.** *Suppose that Assumptions 1, 3 and 4 hold. Let $(x_1, y_1), \ldots, (x_n, y_n) \overset{i.i.d.}{\sim} P(y|x)P(x)$. Let $0 < \delta < 1$. Then the following bound holds with probability at least $1 - \delta$ simultaneously for all $x \in \mathcal{X}$, $k \in \{1, \ldots, n\}$ and $0 < r < r_{\max}$ satisfying the condition (11)*

$$\left\| \Phi(P(\cdot \mid x)) - \mathbb{E}[\Phi(\hat{P}(\cdot \mid x)) \mid X_n] \right\|_{\mathcal{H}} \leq \lambda r \left( \frac{3C_{\mathrm{dist}}k}{nP(B(x,r))} \right)^{1/d}$$

*Proof.* By using the triangle inequality and the Lipschitz continuity of the mapping $x \mapsto \Phi(P(\cdot \mid x))$ in Assumption 1, we obtain

$$\left\| \Phi(P(\cdot \mid x)) - \mathbb{E}[\Phi(\hat{P}(\cdot \mid x)) \mid X_n] \right\|_{\mathcal{H}}$$

$$= \left\| \Phi(P(\cdot \mid x)) - \frac{1}{k} \sum_{j \in \mathrm{NN}(x,k,X_n)} \Phi(P(\cdot \mid x_j)) \right\|_{\mathcal{H}} = \left\| \frac{1}{k} \sum_{j \in \mathrm{NN}(x,k,X_n)} \{\Phi(P(\cdot \mid x)) - \Phi(P(\cdot \mid x_j))\} \right\|_{\mathcal{H}}$$

$$\leq \frac{1}{k} \sum_{j \in \mathrm{NN}(x,k,X_n)} \|\Phi(P(\cdot \mid x)) - \Phi(P(\cdot \mid x_j))\|_{\mathcal{H}} \leq \frac{1}{k} \sum_{j \in \mathrm{NN}(x,k,X_n)} \lambda d_{\mathcal{X}}(x, x_j) \leq \lambda r_{n,k}(x), \tag{19}$$

where $r_{n,k}(x)$ is the distance between $x$ and its $k$-th nearest neighbour in $X_n$. This distance is bounded as in the proof of Kpotufe (2011, Lemma 2), which leads to the claimed bound. For completeness, we prove it here.

The first inequality in the condition (11) implies that

$$a := \frac{k}{n} \geq \frac{\mathcal{V}_\mathcal{B} \ln(2n) + \ln(8/\delta)}{n}.$$

Define a constant $0 < \epsilon < 1$ as

$$\epsilon := \left( \frac{3 C_\text{dist} k}{n P(B(x,r))} \right)^{1/d},$$

where $\epsilon < 1$ follows from the second inequality in the condition (11). Then, Assumption 3 implies that

$$P(B(x, \epsilon r)) \geq C_\text{dist}^{-1} \epsilon^d P(B(x, r)) = 3 \cdot \frac{k}{n} = 3a$$

Thus, Lemma 1 with this choice of $a$ implies that the following holds simultaneously for all $x \in \mathcal{X}$, $k \in \{1, \ldots, n\}$ and $0 < r < r_\text{max}$ satisfying the condition (11) with probability at least $1 - \delta$:

$$P_n\left(B(x, \epsilon r)\right) \geq a = \frac{k}{n} = P_n\left( B(x, r_{k,n}(x)) \right),$$

where the second identity follows from that $r_{k,n}(x)$ is the distance between $x$ and its $k$-nearest neighbour, so the ball of center $x$ and radius $r_{k,n}(x)$ contains $k$ points from $x_1, \ldots, x_n$. This implies that

$$r_{k,n}(x) \leq \epsilon r \leq r \left( \frac{3 C_\text{dist} k}{n P(B(x,r))} \right)^{1/d}$$

simultaneously holds for all $x \in \mathcal{X}$, $k \in \{1, \ldots, n\}$ and $0 < r < r_\text{max}$ satisfying the condition (11) with probability at least $1 - \delta$. The claim is obtained by using this and the bound (19).

$\square$

### A.2 Variance Bound

**Lemma 3.** *Suppose that Assumptions 2 and 4 hold. Let* $(x_1, y_1), \ldots, (x_n, y_n) \overset{i.i.d.}{\sim} P(y|x)P(x)$. *Let* $0 < \delta < 1$. *The following bound simultaneously holds for all* $x \in \mathcal{X}$ *and* $k \in \{1, \ldots, n\}$ *with probability at least* $1 - \delta$:

$$\left\| \mathbb{E}[\Phi(\hat{P}(\cdot \mid x)) \mid X_n] - \Phi(\hat{P}(\cdot \mid x)) \right\|_\mathcal{H}^2 \leq 2 C_\text{ker}^2 \cdot \frac{1 + 4\left(\mathcal{V}_\mathcal{B} \ln(n) - \ln(\delta)\right)}{k}. \tag{20}$$

*Proof.* Denote by $\psi(\text{NN}(x, k, X_n)) \geq 0$ the left hand side of the inequality (20) without the square:

$$\psi(\text{NN}(x, k, X_n)) := \left\| \mathbb{E}[\Phi(\hat{P}(\cdot \mid x)) \mid X_n] - \Phi(\hat{P}(\cdot \mid x)) \right\|_\mathcal{H} \tag{21}$$

$$= \left\| \frac{1}{k} \sum_{j \in \text{NN}(x,k,X_n)} \Phi(P(\cdot \mid x_j)) - \Phi(\hat{P}(\cdot \mid x)) \right\|_\mathcal{H}$$

$$= \left\| \frac{1}{k} \sum_{j \in \text{NN}(x,k,X_n)} \Phi(P(\cdot \mid x_j)) - \Phi(y_j) \right\|_\mathcal{H},$$

where the last expression follows from the definition of $\Phi(\hat{P}(\cdot \mid x))$ in (8). The notation $\psi(\text{NN}(x, k, X_n))$ emphasizes that it depends only on the subset of training data $(x_1, y_1), \ldots, (x_n, y_n)$ associated with the indices $\text{NN}(x, k, X_n)$ of the $k$-nearest neighbours of $x$ in $X_n = \{x_1, \ldots, x_n\}$.

Because of the bound (9), changing $y_i$ for any $i \in \mathrm{NN}(x, k, X_n)$ to any different value $y_i' \in \mathcal{Y}$ changes the value of $\psi(\mathrm{NN}(x, k, X_n))$ at most $2\sqrt{2}C_{\mathrm{ker}}/k$. This can be shown as follows. Let us write the last expression of (21) with the original $y_i$ and the one with $y_i$ replaced by $y_i'$ as

$$\psi(\mathrm{NN}(x, k, X_n))|_{y_i} = \|A + B\|_{\mathcal{H}}, \qquad \psi(\mathrm{NN}(x, k, X_n))|_{y_i'} = \|A' + B\|_{\mathcal{H}},$$

where

$$A := \frac{1}{k}\left(\Phi(P(\cdot \mid x_i)) - \Phi(y_i)\right), \qquad A' := \frac{1}{k}\left(\Phi(P(\cdot \mid x_i)) - \Phi(y_i')\right),$$

$$B := \frac{1}{k}\sum_{j \in \mathrm{NN}(x,k,X_n) \text{ and } j \neq i} \Phi(P(\cdot \mid x_j)) - \Phi(y_j).$$

The triangle inequality implies that

$$\|A + B\|_{\mathcal{H}} \leq \|A\|_{\mathcal{H}} + \|B\|_{\mathcal{H}}, \qquad \|A' + B\|_{\mathcal{H}} \geq \|B\|_{\mathcal{H}} - \|A'\|_{\mathcal{H}}.$$

Therefore,

$$\psi(\mathrm{NN}(x, k, X_n))|_{y_i} - \psi(\mathrm{NN}(x, k, X_n))|_{y_i'} = \|A + B\|_{\mathcal{H}} - \|A' + B\|_{\mathcal{H}}$$

$$\leq \|A\|_{\mathcal{H}} + \|B\|_{\mathcal{H}} - (\|B\|_{\mathcal{H}} - \|A'\|_{\mathcal{H}}) = \|A\|_{\mathcal{H}} + \|A'\|_{\mathcal{H}}.$$

Similarly,

$$\psi(\mathrm{NN}(x, k, X_n))|_{y_i'} - \psi(\mathrm{NN}(x, k, X_n))|_{y_i} \leq \|A\|_{\mathcal{H}} + \|A'\|_{\mathcal{H}}.$$

Hence,

$$\left|\psi(\mathrm{NN}(x, k, X_n))|_{y_i} - \psi(\mathrm{NN}(x, k, X_n))|_{y_i'}\right| \leq \|A\|_{\mathcal{H}} + \|A'\|_{\mathcal{H}}$$

$$= \frac{1}{k}\|\Phi(P(\cdot \mid x_i)) - \Phi(y_i)\|_{\mathcal{H}} + \frac{1}{k}\|\Phi(P(\cdot \mid x_i)) - \Phi(y_i')\|_{\mathcal{H}} \leq \frac{2\sqrt{2}C_{\mathrm{ker}}}{k},$$

where the last inequality follows from the bound (9).

On the other hand, the output $y_i$ associated with any non-$k$-nearest neighbours $i \notin \mathrm{NN}(x, k, X_n)$ does not appear in $\psi(\mathrm{NN}(x, k, X_n))$, so changing the value of $y_i$ in this case does not change $\psi(\mathrm{NN}(x, k, X_n))$.

Thus, for fixed $X_n$, the probability that the random variable $\psi(\mathrm{NN}(x, k, X_n))$ exceeds its expectation $\mathbb{E}[\psi(\mathrm{NN}(x, k, X_n))]$ plus any positive constant $\epsilon > 0$ is upper bounded by using McDiarmid's inequality as

$$\Pr\left(\psi(\mathrm{NN}(x, k, X_n)) > \mathbb{E}\left[\psi(\mathrm{NN}(x, k, X_n)) \mid X_n\right] + \epsilon \mid X_n\right) \leq \exp\left(-\frac{\epsilon^2 k}{4C_{\mathrm{ker}}^2}\right). \tag{22}$$

This is a bound for fixed $x$ and $k$.

Next, for fixed $X_n$, we consider the probability that the statement

$$\psi(\mathrm{NN}(x, k, X_n)) > \mathbb{E}\left[\psi(\mathrm{NN}(x, k, X_n)) \mid X_n\right] + \epsilon \tag{23}$$

holds for *some* $x \in \mathcal{X}$ and $k \in \{1, \ldots, n\}$. The number of *distinct* such statements is identical to the number of distinct index sets of nearest neighbours $\mathrm{NN}(x, k, X_n)$, since the random variable $\psi(\mathrm{NN}(x, k, X_n))$ depends only on the subset of $(x_1, y_1), \ldots, (x_n, y_n)$ associated with $\mathrm{NN}(x, k, X_n)$, as mentioned previously. In other words, if there are other $x' \in \mathcal{X}$ and $k' \in \{1, \ldots, n\}$ that give the identical index set of nearest neighbours as for $x$ and $k$, i.e.,

$$\mathrm{NN}(x', k', X_n) = \mathrm{NN}(x, k, X_n),$$

then the random variable $\psi(\mathrm{NN}(x', k', X_n))$ for $x'$ and $k'$ is identical to that for $x$ and $k$:

$$\psi(\mathrm{NN}(x', k', X_n)) = \psi(\mathrm{NN}(x, k, X_n)).$$

The number of distinct index sets of nearest neighbours is identical to the number of distinct ways the set $X_n$ of $n$ points is intersected by balls $B(x, r_{k,n}(x))$ of center $x$ and radius $r_{k,n}(x)$ being the distance of the $k$-th nearest neighbour from $x$. This number is upper-bounded by the number of distinct ways $X_n$ is intersected by the class $\mathcal{B} = \{B(x, r) \mid x \in \mathcal{X}, \ r > 0\}$ of all balls, which is further upper-bounded by $n^{\mathcal{V}_\mathcal{B}}$ with the VC dimension $\mathcal{V}_\mathcal{B}$ of $\mathcal{B}$ (Kpotufe, 2011, p.6). Therefore, by using the union bound, the probability that the statement (23) holds for some $x$ and $k$ is upper bounded by the bound (22) times $n^{\mathcal{V}_\mathcal{B}}$:

$$\Pr\left(\psi(\mathrm{NN}(x, k, X_n)) > \mathbb{E}[\psi(\mathrm{NN}(x, k, X_n)) \mid X_n] + \epsilon \ \text{ for some } x \in \mathcal{X} \text{ and } k \in \{1, \dots, n\} \mid X_n\right)$$

$$\leq n^{\mathcal{V}_\mathcal{B}} \exp\left(-\frac{\epsilon^2 k}{4 C_{\mathrm{ker}}^2}\right) \quad \text{for all} \quad \epsilon > 0. \tag{24}$$

Now, set

$$\delta = n^{\mathcal{V}_\mathcal{B}} \exp\left(-\frac{\epsilon^2 k}{4 C_{\mathrm{ker}}^2}\right) \iff \epsilon^2 = \frac{4 C_{\mathrm{ker}}^2 \left(\mathcal{V}_\mathcal{B} \ln(n) - \ln(\delta)\right)}{k}.$$

For any value of $0 < \delta < 1$, there is a corresponding $\epsilon > 0$. Then, the bound (24) implies that, for fixed $X_n$, the following upper bound on the random variable $\psi(\mathrm{NN}(x, k, X_n))$ squared holds for *all* $x \in \mathcal{X}$ and $k \in \{1, \dots, n\}$ with at least probability $1 - \delta$:

$$\psi(\mathrm{NN}(x, k, X_n))^2 \leq 2\mathbb{E}[\psi(\mathrm{NN}(x, k, X_n)) \mid X_n]^2 + 2\epsilon^2$$

$$\leq 2\mathbb{E}[\psi(\mathrm{NN}(x, k, X_n))^2 \mid X_n] + 2\epsilon^2,$$

where the second inequality follows from Jensen's inequality. Replacing $\psi(\mathrm{NN}(x, k, X_n))$ by its definition (21) and $\epsilon^2$ by the above expression, we obtain the following bound on the variance term that holds for all $x$ and $k$ with probability at least $1 - \delta$:

$$\left\|\frac{1}{k} \sum_{j \in \mathrm{NN}(x,k,X_n)} \Phi(P(\cdot \mid x_j)) - \Phi(\hat{P}(\cdot \mid x))\right\|_{\mathcal{H}}^2$$

$$\leq 2\mathbb{E}\left[\left\|\frac{1}{k} \sum_{j \in \mathrm{NN}(x,k,X_n)} \Phi(P(\cdot \mid x_j)) - \Phi(\hat{P}(\cdot \mid x))\right\|_{\mathcal{H}}^2 \mid X_n\right] + \frac{8 C_{\mathrm{ker}}^2 \left(\mathcal{V}_\mathcal{B} \ln(n) - \ln(\delta)\right)}{k}. \tag{25}$$

Define $\mathcal{H}$-valued random variables

$$z_j := \Phi(P(\cdot \mid x_j)) - \Phi(y_j) \quad \text{for all} \ \ j \in \mathrm{NN}(x, k, X_n).$$

These random variables are conditionally independent given $X_n$. The conditional expectation of each $z_j$ given $X_n$ is zero, and the conditional variance is uniformly upper bounded due to the bound (9):

$$\mathbb{E}[z_j \mid X_n] = \mathbb{E}[\Phi(P(\cdot \mid x_j)) - \Phi(y_j) \mid X_n] = \Phi(P(\cdot \mid x_j)) - \mathbb{E}[\Phi(y_j) \mid x_j] = 0,$$

$$\mathbb{E}\left[\|z_j\|_{\mathcal{H}}^2 \mid X_n\right] = \mathbb{E}[\|\Phi(P(\cdot \mid x_j)) - \Phi(y_j)\|_{\mathcal{H}}^2 \mid x_j] \leq 2 C_{\mathrm{ker}}^2.$$

Therefore, the first term in the bound (25) can be expressed as (see also the definition of $\Phi(\hat{P}(\cdot \mid x))$ in (8))

$$\mathbb{E}\left[\left\|\frac{1}{k} \sum_{j \in \mathrm{NN}(x,k,X_n)} \Phi(P(\cdot \mid x_j)) - \Phi(y_j)\right\|_{\mathcal{H}}^2 \mid X_n\right] = \mathbb{E}\left[\left\|\frac{1}{k} \sum_{j \in \mathrm{NN}(x,k,X_n)} z_j\right\|_{\mathcal{H}}^2 \mid X_n\right]$$

$$= \mathbb{E}\left[\frac{1}{k^2} \sum_{j \in \mathrm{NN}(x,k,X_n)} \|z_j\|_{\mathcal{H}}^2 + \frac{1}{k^2} \sum_{j \neq m \in \mathrm{NN}(x,k,X_n)} \langle z_j, z_m \rangle_{\mathcal{H}} \mid X_n\right]$$

$$= \frac{1}{k^2} \sum_{j \in \mathrm{NN}(x,k,X_n)} \mathbb{E}\left[\|z_j\|_{\mathcal{H}}^2 \mid X_n\right] + \frac{1}{k^2} \sum_{j \neq m \in \mathrm{NN}(x,k,X_n)} \langle \mathbb{E}[z_j \mid X_n], \mathbb{E}[z_m \mid X_n] \rangle_{\mathcal{H}}$$

$$= \frac{2 C_{\mathrm{ker}}^2}{k}.$$

The proof completes by using this expression in the bound (25) and noting that this bound is independent of $X_n$.

$\square$

