# OpenReview forum: "kNNSampler: Stochastic Imputations for Recovering Missing Value Distributions"
_TMLR — Accepted by TMLR_

### Review · Reviewer_PmYc · 2025-10-15

**Summary Of Contributions:**

The paper proposes a simple kNN-based method for the imputation of missing data. The method, called kNNSampler, estimates the conditional distribution of missing values using a k-nearest neighbor (kNN)–based empirical distribution, rather than imputing single-point estimates such as the mean or median of the kNN samples.

Main Contributions:

Imputation framework: Instead of returning a single deterministic imputed value, kNNSampler approximates the conditional distribution P(y|x) by averaging the responses of the k nearest neighbors. This allows for stochastic imputations that are more realistic when p(y|x) is multimodal, as well as uncertainty quantification for each missing value.

Theoretical convergence analysis: The authors show that the empirical kNN conditional distribution converges to the true conditional distribution in Maximum Mean Discrepancy (MMD) as the sample size n and the number of neighbors k increase suitably. They adapt classical kNN regression convergence proofs to the Hilbert space–valued setting, providing a bias–variance decomposition and rates of convergence under standard smoothness assumptions.

Key Strengths:

Clear presentation: The paper is well-structured and readable, clearly stating the problem, assumptions, and theoretical claims.

Theoretically sound: The convergence analysis is rigorous and grounded in established nonparametric regression theory and RKHS embedding literature.

Reasonable empirical validation: The experiments are consistent with the theoretical claims and demonstrate practical effectiveness on two artificial datasets.

Key Weaknesses:

Motivation for the energy distance used in the experiments: While the energy distance is a valid metric, the paper could better justify why it was chosen over alternatives (e.g., MMD, Wasserstein distance) and discuss its interpretability in the imputation context. For instance, is there a theoretical connection between the energy distance and the Maximum Mean Discrepancy used in the analytical part of the paper?

Limited empirical scope: The experiments are conducted primarily on two artificially generated (toy) datasets. While these are useful for illustrating scenarios in which kNNSampler is more advisable than kNNImputer, they may not fully demonstrate the method’s performance in realistic scenarios. Including results on standard benchmark datasets would provide stronger evidence of the method’s practical applicability.

**Additional Comments:**

No comment

**Audience:**

Yes

**Audience Explanation:**

The paper proposes a simple procedure based on kNN that is more appropriate for multimodal conditional distributions p(y|x) than the classic imputation approach based on the mean or median of the kNN samples, and its convergence is theoretically proved.

**Broader Impact Concerns:**

No concern

**Claims And Evidence:**

Yes

**Claims Explanation:**

The approach is clearly motivated, simple, and supported by a convincing theoretical convergence analysis.

**Requested Changes:**

In the introduction, it would be beneficial to better motivate the stochastic sampling approach compared to the mean/median approach by explaining the differences between regression and imputation. In regression, the goal is typically to minimize the expected loss between the prediction y' and the true (but unknown) value y given x (e.g., squared error). In contrast, in imputation, the objective is to generate samples that follow the true conditional distribution p(y|x). The introduction could start by clearly distinguishing these two goals.

Some claims about the superiority of kNNSampler over kNN x kDE should be moderated. Both methods yield very similar results, and the experiments have been conducted only on two toy datasets.

It is not clear how the parameter k is selected in the experiments for kNNSampler. Is this value different from the one used in kNNImputer (k=5)? Do the authors use the theoretical guidance suggesting k ≈ n^(2/(2+d))?

Figures 5–9 should be replaced with tables, as tables occupy less space and provide more precise numerical information. In any case, Figure 9 should be moved from the references section to the main text.

I would remove the section about p-values, since the results based on the energy distance already provide overlapping and sufficient evidence.

In Section 4.3, the uncertainty quantification analysis should include a similar procedure for kNNImputer, using it as a benchmark for comparison.

---

> ### Author Response · Authors · 2025-11-07
>
> Thank you so much for your helpful comments. Please find below our responses.
>
> > Motivation for the energy distance used in the experiments: While the energy distance is a valid metric, the paper could better justify why it was chosen over alternatives (e.g., MMD, Wasserstein distance) and discuss its interpretability in the imputation context. For instance, is there a theoretical connection between the energy distance and the Maximum Mean Discrepancy used in the analytical part of the paper?
>
> * Thank you for the comment. The explanation of the energy distance in Section 4.1.2 has been expanded to better explain the motivation. The energy distance is a canonical instance of MMD that is both rotation-invariant and scale-invariant (i.e., the distance scales linearly with data scale).
>
> > Limited empirical scope: The experiments are conducted primarily on two artificially generated (toy) datasets. While these are useful for illustrating scenarios in which kNNSampler is more advisable than kNNImputer, they may not fully demonstrate the method’s performance in realistic scenarios. Including results on standard benchmark datasets would provide stronger evidence of the method’s practical applicability.
>
> * Thank you for the suggestion. We conducted a new real-data experiment on solar power data, where missing values are common, and reported the results in a new section (Section 5).
>
> > In the introduction, it would be beneficial to better motivate the stochastic sampling approach compared to the mean/median approach by explaining the differences between regression and imputation. In regression, the goal is typically to minimize the expected loss between the prediction y' and the true (but unknown) value y given x (e.g., squared error). In contrast, in imputation, the objective is to generate samples that follow the true conditional distribution p(y|x). The introduction could start by clearly distinguishing these two goals.}
>
> * Thank you for the suggestion. On this point, new Remarks 1 and 2 have been added to Sections 2.1 and 2.2, which are now referred to in the introduction.
>
> > Some claims about the superiority of kNNSampler over kNN x kDE should be moderated. Both methods yield very similar results, and the experiments have been conducted only on two toy datasets. }
>
> * Thank you for the suggestion. We have moderated the discussion in Section 4.2.2.
>
> > It is not clear how the parameter k is selected in the experiments for kNNSampler. Is this value different from the one used in kNNImputer (k=5)? Do the authors use the theoretical guidance suggesting $k \approx n^{2/(2+d)}$?
>
> * It is determined by leave-one-out cross-validation using a fast exact computation method on the observed covariate-response pairs. This was mentioned in Section 2.3 and is now also explained in Section 4.1.3. Thank you for asking.
>
> >   Figures 5–9 should be replaced with tables, as tables occupy less space and provide more precise numerical information. In any case, Figure 9 should be moved from the references section to the main text.
>
>
> * Thank you for the suggestion. Quantitative results are now presented as tables. Figure 9 (now Figure 5) is now located in the main text. This figure remains as plots; its table version, which we have tried, is less easy to understand.
>
> > I would remove the section about p-values, since the results based on the energy distance already provide overlapping and sufficient evidence.
>
> * The results on p-values have now been removed.
>
> > In Section 4.3, the uncertainty quantification analysis should include a similar procedure for kNNImputer, using it as a benchmark for comparison.
>
> * Since kNNImputer is a deterministic imputation method, it is not straightforward to perform uncertainty quantification with it, and we could not come up with a way to do so within the given time for revision. If you have a specific idea, please let us know.

---

> > ### Comment · Reviewer_PmYc · 2025-11-20
> > **Comments on the responses**
> >
> > > The explanation of the energy distance in Section 4.1.2 has been expanded to better explain the motivation. The energy distance is a canonical instance of MMD that is both rotation-invariant and scale-invariant (i.e., the distance scales linearly with data scale).
> > >> The motivation for using the energy distance is very weakly developed. The text contains many parenthetical remarks that make it difficult to read. The sentence inside the parentheses (“in contrast to the MMD defined with, e.g., a Gaussian kernel, which depends on the bandwidth parameter”) is poorly constructed. It is also unclear why rotation invariance or scale invariance are desirable properties in this context.
> >
> > > We conducted a new real-data experiment on solar power data, where missing values are common, and reported the results in a new section (Section 5).
> >
> > >> The empirical evidence is still insufficient. Only one non-synthetic dataset with two dimensions is included. There are many open-source datasets that could be used. It is not clear why a single two-dimensional dataset is considered, nor why such a sophisticated procedure is used to simulate missingness: “We randomly select a subset of N covariate-response pairs from the full dataset. In this subset, we select randomly 30% of the units whose covariates are between 0.4 and 0.6 and set their responses to missing.”
> >
> > >> If the goal is to study the effect of the training set size, this should be evaluated at different orders of magnitude, e.g., 100,1000, 10000.
> >
> > >>I suggest expanding the experimental section using multiple datasets of different sizes and dimensionalities across several orders of magnitude, and estimating performance via cross-validation, where in the test split one or several values of each instance are removed. One could also choose, for each train–test split, a random feature to be missing for all test instances. At this point, figures are not necessary, and the study should focus on tables with one dataset per row and the results of the different methods.
> >
> > >>The experimental section should be reorganized using new subsections. The current structure is not adequate. I propose: 1.Benchmarks and parameter tuning, performance measure, and estimation of performance, 2.Results on artificial data, 3.Results on real data. Both Sections 2 and 3 should follow a similar structure:
> > – Briefly describe the selected datasets. In Section 2, include a table with the number of dimensions and instances of each dataset. Also include any preprocessing or normalization steps.
> > – Describe how missing values are generated in the test split. For real datasets, keep the missingness generation as simple as possible.
> > – Present the energy distance results. In Section 2, the illustrative figures for the toy datasets are fine, but in Section 3, only tables should be included.
> >
> > >On this point, new Remarks 1 and 2 have been added to Sections 2.1 and 2.2, which are now referred to in the introduction.
> > >>The added remarks do not address my original concern. The introduction should clearly explain the difference between regression and imputation in terms of their objectives. Now that mean squared error results are included for the synthetic datasets, the introduction could note that regression aims to minimize mean squared error, which is optimized by the conditional expectation, but this objective is not appropriate for imputation, where the goal is to recover the full conditional distribution of the missing data given the observed values.
> >
> > >It is determined by leave-one-out cross-validation using a fast exact computation method on the observed covariate-response pairs. This was mentioned in Section 2.3 and is now also explained in Section 4.1.3. Thank you for asking.
> > >>Are the parameters of the other benchmark methods selected using the same procedure? What is the set of candidate values used for the parameters? In the case of random forests, which parameters were tuned and over which ranges?
> >
> > >Quantitative results are now presented as tables. Figure 9 (now Figure 5) is now located in the main text. This figure remains as plots; its table version, which we have tried, is less easy to understand.
> > >>The sample sizes used in the tables are not appropriate, as they should span different orders of magnitude, e.g., 100, 1000, 10000. Replace Figure 5 with a table. Add results for N = 100. It is not clear what the “missing rate” measures, since the intervals become larger for smaller missing rates.
> >
> > >Since kNNImputer is a deterministic imputation method, it is not straightforward to perform uncertainty quantification with it, and we could not come up with a way to do so within the given time for revision. If you have a specific idea, please let us know.
> > >> Please improve the explanation of what coverage probabilities are, what confidence intervals (CIs) represent, and clearly motivate the experiment. For example: What is the objective? How are the intervals generated (e.g., how many imputations are used)?

---

### Review · Reviewer_gEd5 · 2025-10-15

**Summary Of Contributions:**

This paper presents a theoretical analysis of kNNSampler, a data imputation method that fills in missing values by randomly selected a response from the k nearest neighbors based on covariates.

### Contributions

1.	The authors prove weak convergence between the estimated and true conditional distributions of missing values given the covariates. Using maximum mean discrepancy as the distance metric between distributions, they demonstrate that this distance converges to zero when both k/n and ln(n)/k converge to zero, where n is the total number of units used, and k is the number of nearest neighbors.
2.	The proof employs the kernel mean embedding framework, which maps probability distributions to a reproducing kernel Hilbert space (RKHS). The authors adapt the proof by Kpotufe (2011) for real valued kNN regression to their specific imputation context.

3.	The experimental results demonstrate that kNNSampler outperforms commonly used imputation methods in capturing the conditional distributions of mission values when the conditional distributions of the response given the covariates are asymmetric or multi-modal.

### Strengths

1.	Clear high-level message: The paper clearly demonstrates that regression-based imputation methods are inadequate when the goal is to recover the true distribution of missing data through imputation. The figures in the introduction provide digestible visual arguments for why this is the case.
2.	Practical algorithm: The algorithm being analyzed is simple, straightforward algorithm, and can easily be implemented in practice.

### Weaknesses

1.	Proof issue in Lemma 3: see detailed discussion in the next section

**Audience:**

Yes

**Audience Explanation:**

The results of this paper are relevant to the TMLR's audience. The paper provides a theoretical convergence analysis for a simple randomized variation of k-Nearest Neighbor imputation and makes a compelling case for when and why this approach should be used.

**Broader Impact Concerns:**

No concerns

**Claims And Evidence:**

No

**Claims Explanation:**

Overall, the paper provides complete proofs in the appendix and effectively explains the key ideas of the methods and the proof techniques in the main text. The empirical evidence also supports the paper arguments about kNNSampler.

However, I believe there is an error in the proof of Lemma 3, specifically in bounding $|\psi(\text{NN}(x,k,X_n))\mid_{y_i} -\psi(\text{NN}(x,k,X_n))\mid_{y'_i} |$. The issue appears in the final equality where the terms for $j \in \text{NN}(x,k,X_n)$ with $j \neq i$ should be included but appear to be omitted. Please correct me if I’m mistaken. While I believe the statement itself is correct and this error is fixable, it should be addressed in the final version.

**Requested Changes:**

1.	Proof: Correct the error in Lemma 3.
2.	Typos and notation:

    a.	The power of 2/d in equations (10) and (12) is inconsistent.

    b.	 The use of curly brackets in the proof of Lemma 3 might confuse the reader.

3.	Formatting: Figure 9 currently appears in the middle of the references section and should be moved before the references.

---

> ### Author Response · Authors · 2025-11-07
>
> Thank you very much for your detailed comments. Please find our responses below.
>
> >  However, I believe there is an error in the proof of Lemma 3, specifically in bounding ... Please correct me if I’m mistaken. While I believe the statement itself is correct and this error is fixable, it should be addressed in the final version.
>
> * Thank you so much for spotting this error, which is now fixed. The bound remains the same and the rest of the proof works as before. Thanks again!
>
> > The power of 2/d in equations (10) and (12) is inconsistent.
>
> * Thank you for spotting this. It was a typo in (12): the exponent should have been $2/d$ and is now fixed.
>
> > The use of curly brackets in the proof of Lemma 3 might confuse the reader.
>
> * Curly brackets have now been removed. Thank you for the suggestion.
>
> > Formatting: Figure 9 currently appears in the middle of the references section and should be moved before the references.
>
> * Thank you. This has been fixed.

---

### Review · Reviewer_c5Gg · 2025-10-23

**Summary Of Contributions:**

The paper presents kNNSampler, a nearest-neighbour oriented random hot deck data imputation method and conducts a theoretical analysis to establish that it is a principled approach.

The theoretical contributions include an error bound, in terms of maximum mean discrepancy, between the kNN and true conditional distributions for a given, fixed covariate. The bound captures the impact of the number of observed response-covariate pairs (n) and the number of nearest neighbours (k). The bound implies statistical consistency of the kNN conditional distribution, provided that the number of nearest neighbours increases to infinity at a rate slower than n.

The bound is derived by analyzing the mean embedding of the kNN conditional distribution in a RKHS. This can be viewed as a novel estimator for the mean embedding of the true conditional distribution (the conditional mean embedding). The bound provides consistency and convergence rates for the introduced kNN estimator.

The methodology conducts an error analysis on RKHS-valued regression where the response variable is infinite-dimensional. The analysis establishes that the required sample size for a given level of precision increases exponentially with the covariate’s intrinsic dimension, as opposed to its ambient dimension.

**Additional Comments:**

It isn’t obvious to me how the 2/d in (10) becomes d/2 in (12). If this is correct, then I would recommend some text be added to explain the transition.

**Audience:**

Yes

**Audience Explanation:**

Imputation is a very important problem and the kNN-based imputation is widely used, so the findings of the paper would be of use and interest to multiple individuals in TMLR's audience.

**Claims And Evidence:**

Yes

**Claims Explanation:**

The main claims of the paper are theoretical in nature. The theoretical results are based on reasonable assumptions, the theorem statements are clear, and I did not detect any errors in the proofs. These claims are supported by accurate, convincing, and clear evidence.

The remaining claim (in the abstract) is that “Experiments demonstrate its effectiveness in recovering the distribution of missing values”.

The paper provides results for two data settings and compares to relative basic baselines (apart from the kNN x KDE). The support for this claim, as stated, is weak. There are only two investigated datasets, and both are synthetic and low-dimensional. To support this general claim, the experiments need to encompass a much broader range of datasets, pushing to high dimensions and preferably investigating performance in real-data settings where data is removed in various ways. Alternatively, since the primary contribution of the paper is theoretical, the claim could be modified to “illustrate… for low-dimensional synthetic data”. In my view, this wouldn’t diminish the contribution of the paper, because the value is the theory, and the experiments are primarily providing insights and some support.

**Requested Changes:**

The paper makes a strong theoretical contribution and provides some welcome support for kNN-based and hot deck methods. The following changes are requested:

(1)	As discussed above, the final sentence of the paper claims that “Experiments demonstrate its effectiveness in recovering the distribution of missing values.” While the experiments are illustrative and provide useful insights, they are not sufficiently extensive or thorough to support the claim that they “demonstrate [the] effectiveness”. Either the claim should be modified or more comprehensive experimentation should be included. Since this claim only appears in the abstract, and the experiments are not a critical component of the contribution, I would suggest the former – “Experiments illustrate…” or “Experiments provide insights into the behaviour and performance…”

(2)	The second sentence of the paper reads “Imputation — filling in missing values with their estimates — is a common preprocessing step…” Since it is (almost always) a preprocessing step, it is important to keep in mind that what is really important is its effect on the subsequent processing. But the paper largely ignores this, focusing on the output of the pre-processing step. The theoretical results focus on the conditional distribution, with analysis conducted in terms of its mean embedding. While it is natural to assume that the improved matching of the distributions leads to a meaningful improvement in downstream analysis, the paper doesn’t really establish this. There isn’t theory or experiments to quantify the gains that emerge in subsequent regression or estimation of population-based quantities. The paper should be modified to include either a more compelling discussion or to include some theory/experiments that establish the value of targeting the conditional distribution. Lalande and Doya (2023) provide imputation results. The paper would be more compelling if there were results establishing that the imputation quality (either in terms of NRMSE or likelihood) isn’t inferior to kNN x KDE.

(3)	It would be helpful to include a section discussing limitations of the approach. Some of these stem from random hot-deck methods in general and I don’t think they are addressed by the approach. There are settings where suitable neighbours (donors) are not available because of the relationship between the covariates and missingness. In these settings it is possible for parametric (and possibly the KDE) approach to perform better. While these settings are probably ruled out by the missingness and smoothness assumptions for the presented theory, it would be helpful to have a discussion by the authors that clarifies when the technique may prove less effective.

---

> ### Author Response · Authors · 2025-11-07
>
> Thank you so much for your helpful comments. Please find our responses below.
>
> > (1) As discussed above, the final sentence of the paper claims that “Experiments demonstrate its effectiveness in recovering the distribution of missing values.” While the experiments are illustrative and provide useful insights, they are not sufficiently extensive or thorough to support the claim that they “demonstrate [the] effectiveness”. Either the claim should be modified or more comprehensive experimentation should be included. Since this claim only appears in the abstract, and the experiments are not a critical component of the contribution, I would suggest the former – “Experiments illustrate…” or “Experiments provide insights into the behaviour and performance…”
>
> * Thank you for the suggestion, according to which the abstract has been modified.
>
> > (2) The second sentence of the paper reads “Imputation — filling in missing values with their estimates — is a common preprocessing step…” Since it is (almost always) a preprocessing step, it is important to keep in mind that what is really important is its effect on the subsequent processing. But the paper largely ignores this, focusing on the output of the pre-processing step. The theoretical results focus on the conditional distribution, with analysis conducted in terms of its mean embedding. While it is natural to assume that the improved matching of the distributions leads to a meaningful improvement in downstream analysis, the paper doesn’t really establish this. There isn’t theory or experiments to quantify the gains that emerge in subsequent regression or estimation of population-based quantities. The paper should be modified to include either a more compelling discussion or to include some theory/experiments that establish the value of targeting the conditional distribution. Lalande and Doya (2023) provide imputation results. The paper would be more compelling if there were results establishing that the imputation quality (either in terms of NRMSE or likelihood) isn’t inferior to kNN x KDE.
>
> * Thank you so much for your thoughtful comments.
>
> (i)  We have added Remarks 1 and 2 to discuss why estimating the conditional distribution and sampling from it are needed for a subsequent analysis of a quantity of interest.
>
> (ii) We have added results on RMSEs in Section 4. We have also expanded the explanation of the energy distance, which quantifies the quality of imputation in recovering missing-value distributions.
>
> (iii) We have added a discussion in Section 6 on the need to analyze how imputation quality affects the quality of subsequent analyses of a quantity of interest.
>
> >  (3) It would be helpful to include a section discussing limitations of the approach. Some of these stem from random hot-deck methods in general and I don’t think they are addressed by the approach. There are settings where suitable neighbours (donors) are not available because of the relationship between the covariates and missingness. In these settings it is possible for parametric (and possibly the KDE) approach to perform better. While these settings are probably ruled out by the missingness and smoothness assumptions for the presented theory, it would be helpful to have a discussion by the authors that clarifies when the technique may prove less effective.
>
> * Thank you for the suggestion. We have added a discussion on the limitations of our work in Section 6.
>
> > It isn’t obvious to me how the 2/d in (10) becomes d/2 in (12). If this is correct, then I would recommend some text be added to explain the transition.
>
> * Thank you for spotting this. It was a typo in (12): the exponent should have been $2/d$ and is now fixed.

---

### Author Response · Authors · 2025-11-07
**Paper revised**

Dear Reviewers,

Thank you very much again for your positive and helpful comments.
The revised parts in the updated paper are highlighted in blue.
For details, please find individual responses.

Thanks again for your time, and have a good weekend.

Sincerely,
Authors

---

### Author Response · Authors · 2025-11-21

We would like to thank the reviewers and the Action Editor for their constructive feedback.


Regarding the further experimental requests raised by Reviewer PmYc, we would like to clarify our perspective. The primary contribution of the paper is to provide the first theoretical guarantees for kNNSampler, a random hot-deck method, as explained in the Introduction and appreciated by the reviewers. In the revision, we have strengthened the empirical section within this scope by (i) adding a real-data example, (ii) reorganizing the experimental results, (iii) reporting RMSE in addition to the energy distance, and (iv) clarifying the evaluation protocol.


We also note that kNNSampler belongs to the family of hot-deck imputation methods, which are widely used in practice and have been empirically studied in the literature, as explained in the paper. Our goal in this work is therefore not to re-establish their empirical usefulness from scratch, but to provide a first rigorous theoretical analysis of this class of methods.


We fully agree that a broader empirical benchmark across multiple datasets, dimensionalities, and sample sizes would be valuable. However, we feel that conducting a fully fledged large-scale empirical study would be more appropriate for dedicated follow-up work building on the present theoretical results.


In the current paper, the empirical section is intended to illustrate and complement the theoretical analysis, rather than to provide an exhaustive benchmark study, as also noted by Reviewers c5Gg and gEd5. We hope that, when assessed with this scope in mind, the present empirical evaluation is considered reasonable for the intended scope of the submission.


Best wishes,

Authors,

---

### Decision · Action_Editor_Yucq · 2025-11-28

**Recommendation:** Accept as is

**Audience:**

Yes

**Audience Explanation:**

All the three reviwers, as well as myself, agree that the topic is relevant and of interest for the reference community.

**Claims And Evidence:**

Yes

**Claims Explanation:**

The claims and statements made by the authors are well supported by what is presented in the maniscript